# A single day of TGF-β1 exposure activates chondrogenic and hypertrophic differentiation pathways in bone marrow-derived stromal cells

Kathryn Futrega [1,2,3], Pamela G. Robey [1], Travis J. Klein [2], Ross W. Crawford[2] & Michael R. Doran [1,2,3,4,5 ✉]

Virtually all bone marrow-derived stromal cell (BMSC) chondrogenic induction cultures include greater than 2 weeks exposure to transforming growth factor-β (TGF-β), but fail to generate cartilage-like tissue suitable for joint repair. Herein we used a *micro*-pellet model ($5 \times 10^3$ BMSC each) to determine the duration of TGF-β1 exposure required to initiate differentiation machinery, and to characterize the role of intrinsic programming. We found that a single day of TGF-β1 exposure was sufficient to trigger BMSC chondrogenic differentiation and tissue formation, similar to 21 days of TGF-β1 exposure. Despite cessation of TGF-β1 exposure following 24 hours, intrinsic programming mediated further chondrogenic and hypertrophic BMSC differentiation. These important behaviors are obfuscated by diffusion gradients and heterogeneity in commonly used *macro*-pellet models ($2 \times 10^5$ BMSC each). Use of more homogenous *micro*-pellet models will enable identification of the critical differentiation cues required, likely in the first 24-hours, to generate high quality cartilage-like tissue from BMSC.

[1] National Institute of Dental and Craniofacial Research (NIDCR), National Institutes of Health (NIH), Department of Health and Human Services, Bethesda, MD, USA. [2] Centre for Biomedical Technologies (CBT), Queensland University of Technology (QUT), Brisbane, Queensland, Australia. [3] Translational Research Institute (TRI), Brisbane, Queensland, Australia. [4] School of Biomedical Sciences, Faculty of Health, Queensland University of Technology (QUT), Brisbane, Queensland, Australia. [5] Mater Research Institute, University of Queensland (UQ), Brisbane, Queensland, Australia. ✉email: michael.doran@qut.edu.au

Bone marrow-derived stromal cells (BMSCs, also known as bone marrow-derived "mesenchymal stem cells") were heralded as a panacea for articular cartilage repair, but have failed to live up to expectations[1]. While BMSCs appear to have the capacity to differentiate into chondrocytes, current protocols yield temporary, unstable cell populations that evolve to form hypertrophic chondrocytes and mineralized tissue in vivo[1,2]. Because cartilage is matrix-rich (98%)[3], and matrix underpins mechanical function, chondrogenic assays are biased toward the use of matrix characterization as a measure of BMSC differentiation. However, matrix accumulation takes time, and thus matrix accumulated by the culture endpoint likely reflects cell fate decisions made days earlier. This temporal complexity is frequently exasperated by the use of macroscopic cartilage tissue models that suffer from profound diffusion gradients, yielding both heterogeneous matrix and cell phenotypes[4,5].

Examination of the classic "pellet culture" highlights how temporal and spatial heterogeneity can confound the study of BMSC chondrogenesis. In 1998, Johnstone et al. described the pellet culture (hereafter, referred to as the macro-pellet), which has become the gold standard for differentiating BMSCs into chondrocyte-like cells in vitro[6]. In the original macro-pellet culture, ~$2 \times 10^5$ BMSCs were pelleted in induction medium that critically included supplementation with transforming growth factor-β1 (TGF-β1). Thousands of papers have since used the macro-pellet culture to characterize or optimize the in vitro chondrogenic capacity of BMSCs, or stromal cells from other tissues. Little has changed since the introduction of the macro-pellet culture model, with most studies continuing to use similar numbers of cells per macro-pellet, and several weeks of culture in induction medium supplemented with one of the TGF-β isoforms (TGF-β1, 2, or 3). Recent reviews of published chondrogenic induction models described the use of TGF-β induction ranging from 7 days to 28 days, with macro-pellets containing between $2 \times 10^5$–$1 \times 10^6$ BMSCs each[7,8]. Other growth factors have been incorporated to enhance chondrogenic induction protocols[7–11], mostly in macro-pellet cultures formed from >$2 \times 10^5$ BMSCs each and induction periods ranging from 7–45 days; however, TGF-β remains the most important and most commonly used chondrogenic induction factor. While considerable parallel investment has been made into the development of scaffolds/gels for BMSC cartilage repair, most cultures rely on biological understanding derived from macro-pellet cultures, and use TGF-β supplemented medium formulations similar to that originally described in 1998[6].

TGF-β instructs BMSCs to take on a chondrocyte-like phenotype, and in response, pelleted cells secrete cartilage-like matrix. Macro-pellet cultures evolve to form large (1–3 mm diameter) tissues, yielding large radial diffusion gradients and the development of different cell and tissue types in different regions of the pellet[4,5]. The radially heterogeneous matrix distribution that can be observed at different time points suggests that some regions of the macro-pellet undergo differentiation at different rates than other regions of the macro-pellet[5]. While spatial heterogeneity is well-recognized, the potential impact that this imposes on temporal differentiation heterogeneity is rarely discussed. As BMSC chondrogenic differentiation lies on a continuum, where the terminally differentiated cell type is a hypertrophic chondrocyte[12], the temporal response to induction factors is a critical variable. Geometric heterogeneity, common in macro-pellet cultures, and other macroscopic tissue models, effectively imposes temporal heterogeneity within a single tissue. Because of these uncontrolled gradients, fundamental understanding of BMSC induction kinetics and temporal response to factors such as TGF-β1 remain unknown. Lack of understanding of induction kinetics has hindered development of protocols that might better control stage-wise differentiation of BMSCs into a cell population capable of producing hyaline cartilage, with reduced propensity to undergo hypertrophic differentiation.

Here, we use a more homogeneous, small diameter micro-pellet model to characterize the temporal influence of TGF-β1, the most commonly used BMSC chondrogenic induction factor. We used a high-throughput microwell platform, termed the Microwell-mesh[5], to enable manufacture of thousands of small diameter micro-pellets for this analysis. Because individual micro-pellets ($5 \times 10^3$ cells each) are formed from fewer cells than macro-pellets ($2 \times 10^5$ cells each), the resulting smaller tissues experience reduced diffusion gradients, generally leading to the more uniform supply of both metabolites and signal molecules. The resulting, more homogeneous, cartilage-like tissues are better suited for studying BMSC chondrogenic differentiation kinetics. To account for the disconnect or lag time between cellular response to TGF-β1 exposure and cartilage-like matrix accumulation, we used 21-day cultures, where cells were exposed to TGF-β1 for different time periods ranging from 0, 1, 3, 7, 14, or 21 days. We contrasted results from micro-pellet and macro-pellet tissues formed from either BMSCs or in vitro expanded articular chondrocytes (ACh). We performed RNA sequencing of micro-pellet cultures at various days (0, 1, 3, 7, and 21) and on replicate 21-day cultures where the medium had only been supplemented with TGF-β1 for the first 1, 3, or 7 days. We observed that when diffusion gradients were minimized using the micro-pellet model, chondrogenic and hypertrophic differentiation cascades were triggered in BMSCs with as little as a single day of TGF-β1 exposure. By contrast, in macro-pellets, BMSC differentiation appeared to be progressive, which could lead to the interpretation that extended TGF-β1 exposure is required for BMSC induction. In micro-pellets, intrinsic signaling propagated a differentiation program that resulted in a similar pattern of gene expression in BMSC exposed to TGF-β1 for 1, 3, 7, or 21 days. While it is routine to optimize BMSC chondrogenic induction over multi-week cultures[13–15], our data suggest it may be more useful to focus optimization efforts on the first few hours or days of induction culture. Overall, micro-pellet studies reveal the rapid temporal response of BMSCs to TGF-β1 exposure, the influence of intrinsic programming following TGF-β1 withdrawal, the confounding influence macro-pellet heterogeneity has on perceived differentiation kinetics, and the divergent response of ACh and BMSCs to TGF-β1.

## Results

**Micro- and Macro-pellet culture establishment.** We assembled traditional macro-pellet cultures from $2 \times 10^5$ BMSCs each in deep-well plates (Fig. 1a), while micro-pellets were assembled from $5 \times 10^3$ BMSCs each (40-fold fewer cells per tissue) in Microwell-mesh plates (Fig. 1b). In Microwell-mesh plates, cells in suspension were centrifuged through the openings (36 μm) of the nylon meshes bonded over microwells. Cells pelleted in microwells aggregated to form micro-pellets, becoming too large to escape back through the mesh openings, thus being retained in discrete microwells over the culture period. Larger diameter macro-pellets inherently suffer from increased diffusion gradients of metabolites, gases, and other factors (Fig. 1c)[16,17], while gradients are reduced in smaller diameter micro-pellets, yielding more homogeneous cartilage-like tissues[4,5]. Induction cultures consisted of basal chondrogenic medium supplemented with TGF-β1 for 0, 1, 3, 7, 14, or 21 days of the total 21-day induction culture. Each culture condition is represented by a horizontal line in Fig. 1d, with blue and gray lines representing days with (+)TGF-β1 and without (−)TGF-β1, respectively, and the red arrow specifying the day of analysis. On the indicated day,

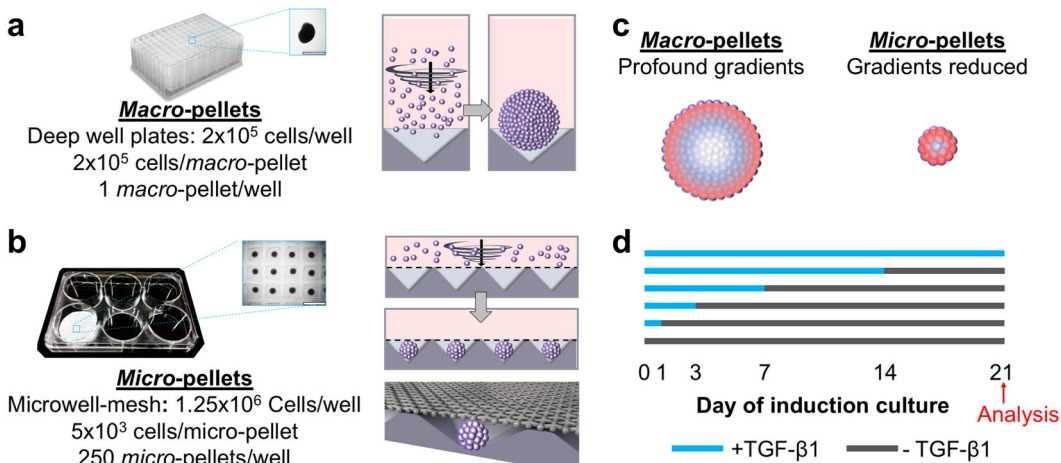

**Fig. 1 Schematic of experimental procedures. a, b** Different cell seeding densities were used to generate *macro*-pellets or *micro*-pellets of specific size in deep-well plates or Microwell-mesh plates, respectively. *Micro*-pellets were retained in discrete microwells by the nylon mesh bonded over the microwell openings (see Supplementary Movie 1). Retention by the mesh enabled long-term *micro*-pellet culture, including multiple medium exchanges. **c** Diffusion gradients are reduced in smaller diameter *micro*-pellets relative to larger diameter *macro*-pellets[16,17]. **d** Cultures were carried out for 21 days total, with blue lines representing culture days with TGF-β1 in the medium, and gray lines representing culture days without TGF-β1 in the medium.

TGF-β1 was washed away and replaced with basal medium for the remainder of the 21-day culture. Parallel cultures were established using BMSCs from four unique donors and ACh from two unique donors.

**Gross morphology, histology, and biochemical content.** Both BMSC *macro*-pellet and *micro*-pellet cultures that were not exposed to TGF-β1 during the culture period remained small (Fig. 2a) and had little matrix glycosaminoglycan (GAG) staining (Toluidine blue, Fig. 2b). *Macro*-pellets grew incrementally larger with extended TGF-β1 exposure (Fig. 2a), while *micro*-pellets achieved the majority of tissue growth in response to a single day of TGF-β1 exposure. Following 21 days of TGF-β1 exposure, *macro*- and *micro*-pellets increased in diameter by 2.17 ± 0.05-fold and 3.69 ± 0.46-fold, respectively, yielding an average tissue volume ~25% greater per input cell in *micro*-pellet cultures. BMSC *macro*-pellets exposed to TGF-β1 for less than 21 days, simultaneously, had regions of undifferentiated tissue (weak GAG staining with Toluidine blue) and regions of differentiated tissue (strong GAG staining, Fig. 2b). By contrast, BMSC *micro*-pellets had relatively uniform and intense GAG staining following a single day of TGF-β1 exposure (Fig. 2b). Substantial BMSC donor variability was observed in *macro*-pellets when TGF-β1 exposure was reduced (<14 days), while all BMSC donor populations yielded intense GAG staining in *micro*-pellets with a single day of TGF-β1 exposure (Fig. 2b). In BMSC *macro*-pellet cultures, maximum GAG content was reached with 7 days of TGF-β1 exposure for one of the four BMSC donors, 14 days of TGF-β1 exposure for two BMSC donors, while one donor required the full 21 days of TGF-β1 exposure (Fig. 2c). In BMSC *micro*-pellet cultures, three of four BMSC donors reached maximum GAG content with 3 days of TGF-β1 exposure, while one donor reached the maximum value following 1 day of TGF-β1 exposure. Similarly, *micro*-pellets reached maximum DNA and GAG/DNA content with less TGF-β1 exposure than required by *macro*-pellets.

ACh *macro*-pellets benefited from extended TGF-β1 exposure, gradually increasing in size (Fig. 3a) and GAG content (Fig. 3b, c). Unlike BMSC *micro*-pellets, ACh *micro*-pellets required prolonged TGF-β1 exposure for maximal growth (Fig. 3a), and GAG production. ACh *macro*- and *micro*-pellets demonstrated reduced GAG staining (Fig. 3b) and content (Fig. 3c) when

TGF-β1 was withdrawn early in the 21-day culture period. Unlike spherical BMSC-derived tissues, ACh *macro*-pellets exhibited a cup-like structure with a concave center (see red arrows pointing to side views in Fig. 3a). This cup-like structure was not observed in ACh *micro*-pellets, which were spherical, like BMSC *micro*-pellets (Fig. 2a).

**qPCR analysis.** We assessed expression of genes associated with chondrogenesis (*SOX9*, *COL2A1*, and *ACAN*) and hypertrophy (*COL1A1*, *COL10A1*, and *RUNX2*) for both *macro*- (Fig. 4a) and *micro*-pellet (Fig. 4b) cultures. While in BMSC *macro*-pellets both chondrogenic and hypertrophic genes tended to increase with extended TGF-β1 exposure, these were not statistically significant for all donors and at all time points analyzed. *Micro*-pellet *COL2A1* and *ACAN* expression was comparable to the 21-day control condition with 1 day of TGF-β1 exposure for BMSC donors 3 and 4, and 3 days for BMSC donors 1 and 2. The expression of *COL10A1* in *micro*-pellets formed from BMSC donors 3 and 4 reached 21-day control levels with 1 day of TGF-β1 exposure, while *micro*-pellets formed from BMSC donor 2 required 3 days of TGF-β1 exposure, and BMSC donor 1 required 7 days of TGF-β1 exposure. *COL1A1* expression in BMSC *micro*-pellets was the only analyzed gene that declined significantly when TGF-β1 was washed out before day 21 for all donors. For ACh, *ACAN* expression declined when TGF-β1 was washed out at days 1, 3, 7, or 14, but this was only significant for one of the two ACh donors, ACh 1. No hypertrophic gene expression was observed in cultures from either of the two ACh donors. Overall, qPCR analysis suggested that 1–3 days of BMSC *micro*-pellet exposure to TGF-β1 was sufficient to induce chondrogenic genes, and this same brief TGF-β1 exposure appeared to also upregulate hypertrophy genes, similar to the 21-day control cultures.

**Assessment of hypertrophy in vivo.** *Micro*-pellets were packed into bovine osteochondral defect models and implanted sub-cutaneously in NOD-scid IL2R gamma[null] (NSG) mice[18] for 8 weeks using a previously established method[19]. Histological examination of harvested tissues showed that, unlike ACh-derived *micro*-pellets, BMSC *micro*-pellets formed mineralized tissue in vivo regardless of TGF-β1 exposure duration (Fig. 5). Hypertrophy was evident from the presence of bone-like tissue in histological samples and Micro-CT analysis indicating a tissue

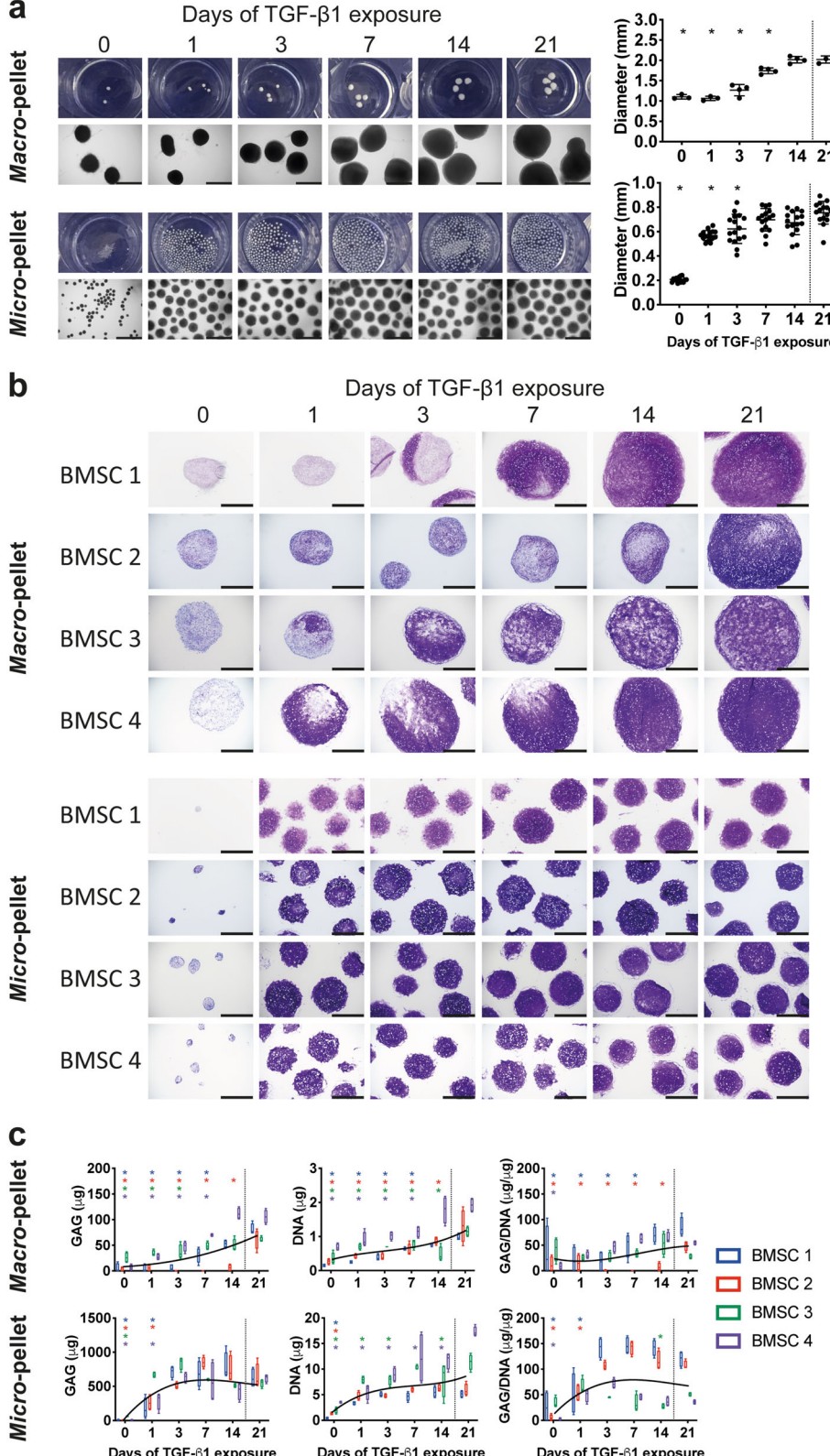

**Fig. 2 Cartilage-like tissues derived from BMSCs after 21 days in vitro differentiation with TGF-β1 exposure for the number of days indicated. a** Gross photographs and brightfield microscopy images of whole BMSC-derived tissues (scale bars for microscopy images, 1 mm), and corresponding diameters. **b** Toluidine blue stain of glycosaminoglycan (GAG) matrix in *macro-* and *micro*-pellet tissue cross-sections are shown for four unique BMSC donors (scale bars, 500 μm). **c** GAG, DNA, and normalized GAG/DNA quantification for each BMSC donor. Asterisks are shown for values that are significantly lower than the corresponding 21 days of TGF-β1 exposure control. Box plots: $n = 4$ for each donor; *$P < 0.05$.

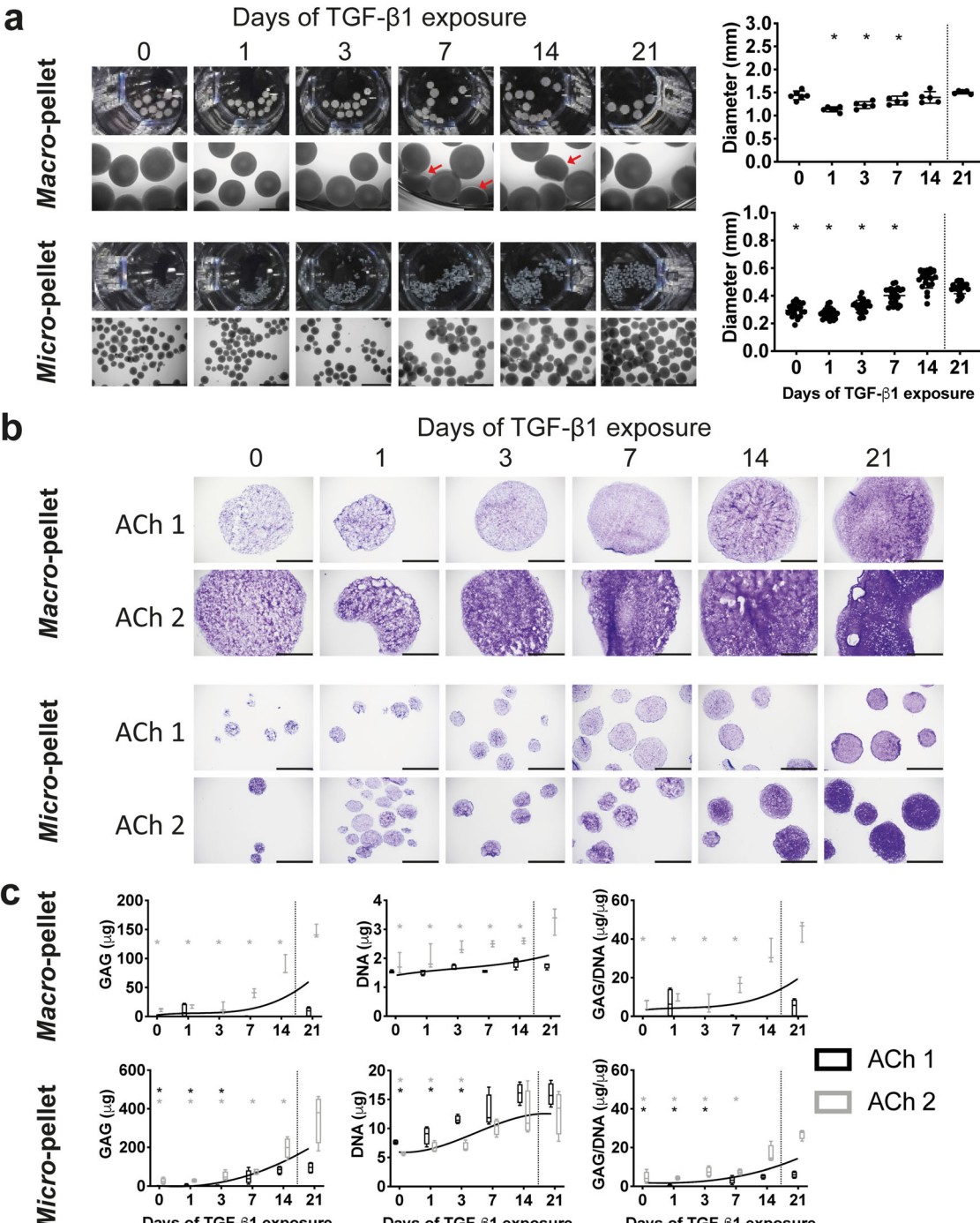

**Fig. 3 Cartilage-like tissues derived from ACh after 21 days in vitro differentiation with TGF-β1 exposure for the number of days indicated. a** Gross photographs and brightfield microscopy images of whole ACh-derived tissues (scale bars for microscopy images, 1 mm), and corresponding diameters. **b** Toluidine blue stain of *macro*- and *micro*-pellet tissue cross-sections are shown for two unique ACh donors (scale bars, 500 µm). **c** GAG, DNA, and normalized GAG/DNA quantification for ACh donors. Asterisks are shown for values that are significantly lower than the corresponding 21 days of TGF-β1 exposure control. Box plots: n = 4 for each donor; *P < 0.05.

density similar to that of the bovine bone in the artificial cartilage defect model (Fig. 5a). Previous studies show that when BMSC pellets, that have been exposed to TGF-β for 3–7 weeks in vitro, are implanted ectopically in mice, that they are iteratively remodeled[20–22]. This remodeling results in mineralization and progressive replacement of cartilage-like tissue with bone or bone marrow-like tissue. Our data demonstrate that BMSC *micro*-pellets exposed to as little as a single day of TGF-β1 mineralized

in vivo. Remodeling of *micro*-pellets at the periphery was obvious, and in some cases, small pockets of marrow could be observed (evident between BMSC *micro*-pellets exposed to TGF-β1 for 21 days in Fig. 5a). In contrast to BMSC *micro*-pellets, ACh *micro*-pellets did not mineralize in vivo (Fig. 5b). GAG staining of ACh *micro*-pellets was faint, suggesting that the cartilage-like tissue may be of low quality, possibly due to the high passage number (passage 6) of the ACh donor cells.

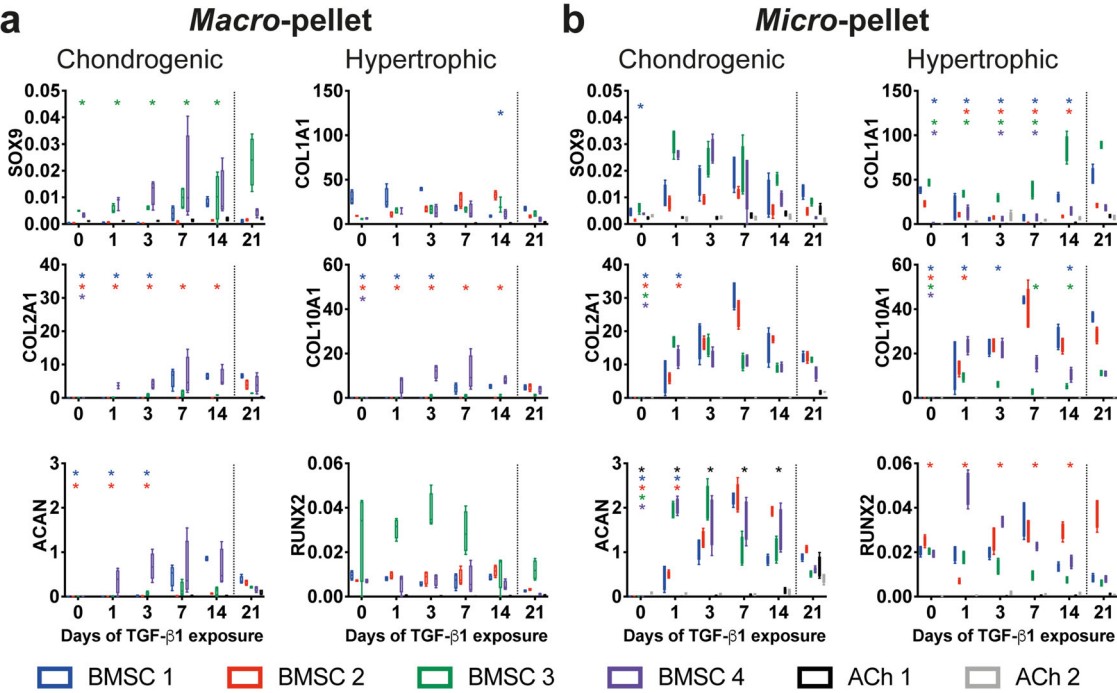

**Fig. 4 qPCR analysis of cartilage-like tissues derived from BMSCs or ACh on day 21 of culture, following varying days of TGF-β1 exposure. a** qPCR from *macro*-pellet cultures, **b** qPCR from *micro*-pellet cultures. All values are compared with the standard 21 days of TGF-β1 exposure control (separated by the dotted line). Gene expression values represent $2^{-\Delta Ct}$, normalized to *GAPDH*. Asterisks are shown for values that are significantly lower than the corresponding 21 days of TGF-β1 exposure control. Box plots: $n = 4$ for each donor; *$P < 0.05$.

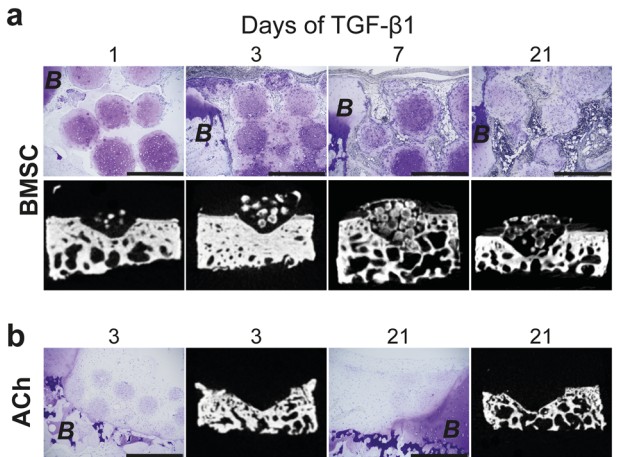

**Fig. 5 *Micro*-pellet tissue formation in vivo.** Analysis of *micro*-pellets formed from **a** BMSCs or **b** ACh implanted subcutaneously in NSG mice for 8 weeks in bovine defect models. Histological sections showed that BMSC and ACh *micro*-pellets appeared highly remodeled with reduced GAG staining (Toluidine blue), and only BMSCs formed bone-like tissues (scale bars, 500 μm). Micro-CT analysis confirmed the presence of mineralized tissue in BMSC *micro*-pellets, unlike ACh *micro*-pellets. Representative images are shown for 21 day in vitro differentiated tissues with TGF-β1 exposure for days indicated above the images. Surrounding bovine cartilage and bone tissue is marked with the letter "*B*" to indicate bovine tissue.

**Cell fate and RNA-seq analysis**. To resolve globally modified gene expression by TGF-β1 between BMSC and ACh cultures, and to identify potential targets to inhibit hypertrophy, we performed bulk RNA-seq analysis. A schematic of the experiment is shown in Fig. 6a with blue and gray lines representing culture days with (+)TGF-β1 and without (−)TGF-β1, respectively, and red arrows representing the day of analysis. We analyzed cultures

before induction (day 0) and harvested on days 1, 3, 7, and 21 from cultures continuously exposed to TGF-β1. Additionally, we analyzed *micro*-pellets cultured until day 21, with exposure to TGF-β1 for the first 1, 3, or 7 days of the total 21-day culture.

As shown in the multi-dimensional scaling (MDS) plot in Fig. 6b, the three BMSC donors analyzed on a specific day after TGF-β1 treatment clustered well with each other at each time point. With incremental culture in TGF-β1-supplemented medium, BMSC gene expression shifted to the upper left corner of the plot for all BMSC donors. When 21-day cultures were replicated with BMSC donor 1 (open blue points in Fig. 6b), but with TGF-β1 eliminated from culture medium at day 1, 3, or 7, gene expression patterns converged (see green shaded oval in Fig. 6b), and overlapped with 21 day cultures that received continuous TGF-β1 supplementation. This clustering of day 21 cultures, regardless of the length of TGF-β1 exposure, validated that BMSCs were fated to a common chondrogenic and hypertrophic differentiation program. Importantly, BMSCs and ACh did not converge in the MDS plot, showing dissimilarity in gene expression, despite being kept under identical culture conditions. While ACh that had been exposed to fewer days of TGF-β1 (open gray points in Fig. 6b) clustered closely together on day 21, they did not completely overlap with ACh that had been exposed to TGF-β1 for the full 21-day culture duration. This suggests that, unlike BMSCs, ACh might undergo de-differentiation, or other transcriptional program, in response to the withdrawal of TGF-β1.

At each of the timepoints that we performed RNA-seq, we analyzed the data for differentially expressed genes between BMSCs and ACh (see Supplementary Data 1). We compiled a list of genes that were consistently differentially expressed (>2 logfold) between BMSC and ACh samples on days 3, 7, and 21, which resulted in a list of 947 genes. We assessed this list of genes against gene ontology terms related to cartilage development (GO:0051216) and ossification (GO:0001503)[23], reducing the list

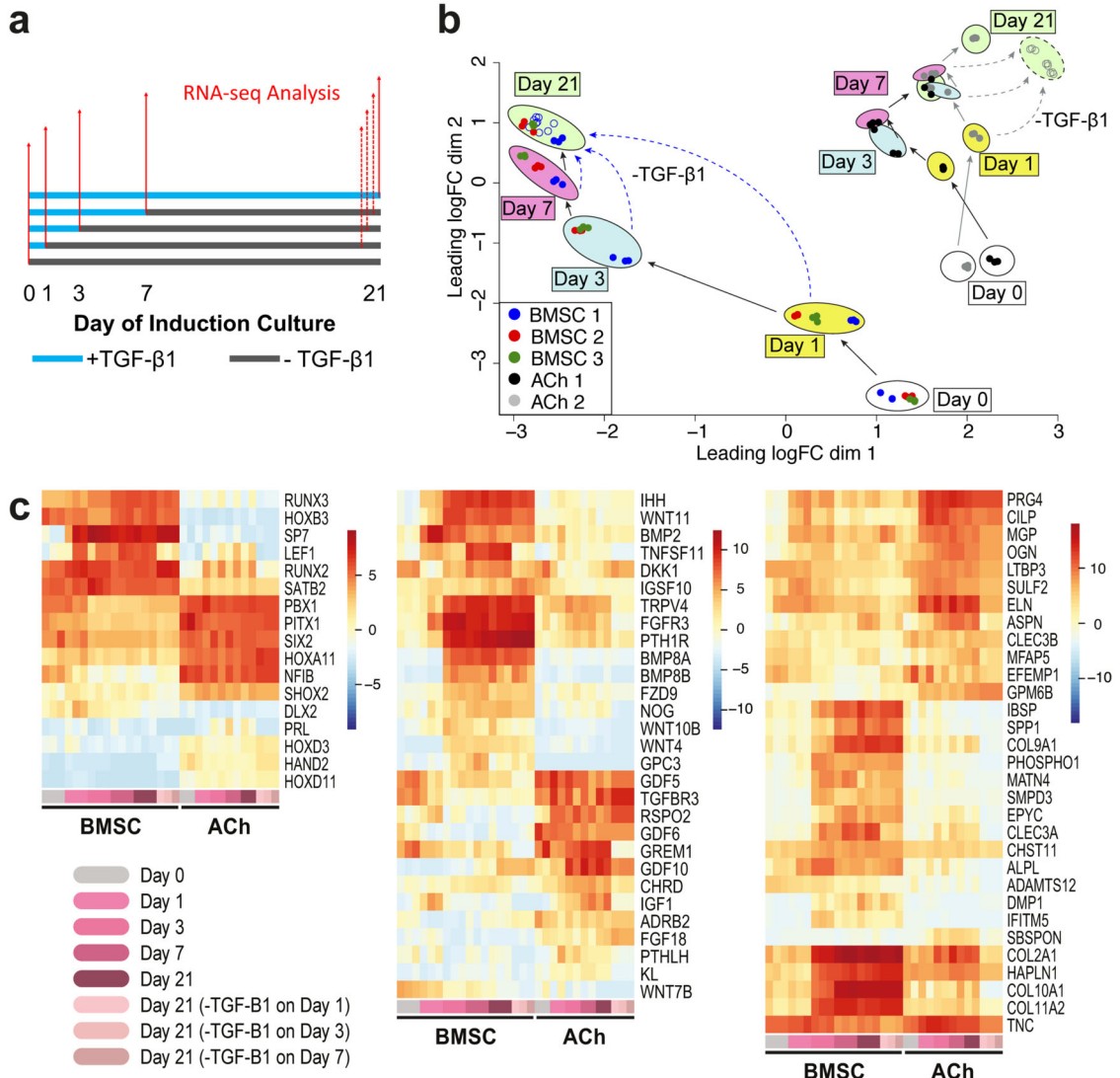

**Fig. 6 RNA-sequencing analysis. a** Whole transcriptome analysis was performed for BMSC and ACh cultures treated with TGF-β1 (blue lines) and harvested on days 0, 1, 3, 7, and 21 as indicated by the solid red arrows (BMSC donors 1, 2, and 3, and ACh donor 1 and 2). Additionally, for BMSC donor 1 and ACh donor 2, RNA-seq analysis was performed on day 21 following TGF-β1 withdrawal on days 1, 3, and 7, represented by dashed red arrows. **b** MDS plot reveals the convergence of BMSC cultures on a common gene expression profile by day 21. This gene expression profile was common in 21-day cultures, despite being exposed to exogenous TGF-β1 for different durations (1, 3, 7, or 21 days). BMSC and ACh gene expression was dissimilar in samples treated in the same way. Solid arrows in the MDS plot show the progression of gene expression profiles from day 0 cultures maintained continuously in TGF-β1, and analyzed on day 1, 3, 7, or 21. Dashed arrows point to the changes in global gene expression on day 21, following TGF-β1 withdrawal from cultures on day 1, 3, or 7. **c** Differentially expressed genes related to osteochondral transcription factors (left), soluble factors and receptor signaling (middle), and ECM molecules and ECM biosynthesis (right). Each timepoint in the heatmap is represented by a color shown on the bottom row of the heatmap, which corresponds to the color key on the bottom left of the figure. Each BMSC timepoint has three columns, representing three unique BMSC donors, and each ACh timepoint has two columns, representing two unique ACh donors; only one BMSC donor and one ACh donor was sequenced on day 21 with TGF-β1 withdrawal on day 1, 3, or 7, and are represented by the lighter pink shades; heatmap colors represent the average of normalized logCPM values from three culture wells.

to 77 relevant genes. We further categorized this list into genes related to transcription factors, soluble factors and receptor signaling, and extracellular matrix (ECM) molecules and ECM biosynthesis, and plotted the gene logCPM values for each sample to generate heatmaps (Fig. 6c).

Genes for transcription factors that were expressed higher in BMSCs, compared with ACh, included *RUNX3, HOXB3, SP7, LEF1, RUNX2*, and *SATB2*. Of these, *SP7* and *LEF1* were specifically upregulated in response to TGF-β1 exposure in BMSCs. However, unlike *LEF1*, *SP7* remained highly expressed on day 21 even with the withdrawal of TGF-β1 on day 1, 3, or 7. A number of genes coding for transcription factors were

significantly higher in ACh, compared with BMSCs, but these genes did not appear to change exclusively in response to TGF-β1 or its withdrawal. Many of the transcription factors that were expressed higher in ACh, including a number of *HOX* genes, *PBX1*[24], *PITX1*[25], *SIX2*[26], *NFIB*[27], and *SHOX2*[28], have been associated with patterning and early musculoskeletal progenitor cells.

Genes associated with soluble signaling factors and receptors that were differentially expressed between BMSCs and ACh were largely related to TGF-β superfamily signaling, the WNT signaling pathway, and a few others. *IHH, WNT11, WNT4*[29], and *BMP2*[30] are associated with cartilage hypertrophy, and in our

study they were upregulated by a single day of TGF-β1 exposure in BMSCs, and their expression remained elevated following its withdrawal. *PTH1R* expression, which is associated with bone formation and resorption[31], was expressed significantly higher in BMSCs compared with ACh, in response to any duration of TGF-β1 exposure analyzed. ACh expressed higher levels of *PTHLH*, a known modulator of cartilage hypertrophy[32], than BMSC, in the presence of TGF-β1, but not following its withdrawal. Chondrocyte-associated factors from the TGF-β superfamily, *GDF5, GDF6*[33], and *TGFBR3*[34], were consistently expressed in ACh, but were underexpressed by BMSCs exposed to TGF-β1. BMP antagonists *GREM1* and *CHRD* were expressed by both BMSCs and ACh prior to induction (day 0) and in the presence of TGF-β1, with higher relative expression in ACh, and decreased in both cell types following TGF-β1 withdrawal. *FGF-18*, a mitigator of hypertrophy[35], was expressed higher in ACh than BMSCs at each timepoint and regardless of TGF-β1 exposure or following its withdrawal.

In both BMSCs and ACh, the expression of genes related to ECM molecules and ECM biosynthesis that were responsive to TGF-β1 were typically upregulated within either 1 day, or by 3 days, of TGF-β1 exposure, and generally remained upregulated following TGF-β1 withdrawal with some exceptions in ACh. In both BMSCs and ACh, *PRG4* and *CILP* were upregulated after 1 day of TGF-β1 exposure, with overall greater expression in ACh, even following TGF-β1 withdrawal. In BMSCs, but not ACh, genes associated with ECM mineralization were upregulated including *IBSP, SPP1*, and *PHOSPHO1*, but required more than 1 day of TGF-β1 exposure and remained highly expressed following its withdrawal. The upregulation of ECM molecule genes associated with cartilage and bone development including *COL2A1, HAPLN1, COL10A1*, and *COL11A2* required more than 1 day of TGF-β1 exposure in BMSCs, but remained highly expressed despite TGF-β1 withdrawal. While *COL2A1* and *ELN* expression in ACh was generally increased with prolonged TGF-β1 expression, TGF-β1 withdrawal significantly reduced the level of their expression. A low expression of the hypertrophic ECM molecule gene *COL10A1* was detected in ACh prior to induction and during TGF-β1 exposure, but this expression dropped dramatically following TGF-β1 withdrawal.

## Discussion
We reason that significant knowledge gaps in the understanding of BMSC chondrogenic differentiation play a major role in the failure of BMSCs to meet clinical cartilage repair expectations[1,36]. Since the late 1990s, researchers have been using the pellet culture[6] (referred to as the *macro*-pellet model in this publication), in which several hundred-thousand BMSCs are pelleted in a tube containing medium supplemented with TGF-β to study this differentiation process. Pelleted BMSCs are a logical input into the chondrogenic differentiation process. During development, mesenchymal stem cell aggregation (condensation) is requisite for cartilage formation[37]. In vitro, TGF-β1 facilitates BMSC chondrogenic differentiation in part through the indirect downregulation of N-cadherin, causing relaxation of cytoskeletal tension, leading to reduced RhoA/ROCK signaling and upregulation of chondrogenic signaling[38]. While cumulatively, these studies suggest that pelleted BMSCs are a logical input into the chondrogenic differentiation process, they do not indicate how pellet size might influence outcome.

The field has recently focused on obstructing the propensity of BMSCs to form hypertrophic tissue[1]. For example, studies report successful obstruction of hypertrophy and formation of stable articular cartilage-like tissue by: (1) manipulation of WNT signaling[13,14]; (2) silencing of BMP receptor signaling via a proprietary molecule[15]; or (3) use of novel scaffolds[39]. While promising, a recent article critically notes that these studies have yet to be replicated by independent groups[40]. A common feature of these previous studies was the use of *macro*-pellet models and extended TGF-β exposure. For example, Narcisi et al. used $2 \times 10^5$ BMSCs per *macro*-pellet and 5 weeks induction (10 ng/mL TGF-β1)[13]; Yang et al. $2.5 \times 10^5$ BMSC per *macro*-pellet and 14–35 days induction (5 ng/mL TGF-β3)[14]; and Occhetta et al. $2.5 \times 10^5$ BMSC per *macro*-pellet and 14 days induction (10 ng/mL TGF-β3)[15]. Physically large tissues, generated from hundreds of thousands of cells, suffer diffusion gradients, resulting in spatial tissue heterogeneity, likely to obfuscate the effects of the differentiation media. Stable cartilage-like tissue may have formed in regions of these *macro*-tissues, but spatially varied tissue quality may confound further optimization.

Our group has been a proponent of using smaller diameter *micro*-pellets to generate more homogeneous tissues[4,5], reasoning that a homogeneous readout is necessary for optimizing a complex bioprocess such as BMSC chondrogenesis. Here we studied BMSC differentiation in response to TGF-β1 using a *micro*-pellet model ($5 \times 10^3$ BMSCs each) and a traditional *macro*-pellet model ($2 \times 10^5$ BMSCs each). Physical tissue size provided an immediate indication that the kinetics of BMSC differentiation in response to TGF-β1 exposure differed in *micro*- and *macro*-pellets. As expected, extended TGF-β1 exposure resulted in incrementally larger BMSC *macro*-pellets with greater GAG content[5,41]. Unexpectedly, a single day of TGF-β1 exposure yielded BMSC *micro*-pellets that grew in size and accumulated GAG quantity nearly equivalent to *micro*-pellets exposed to TGF-β1 for 21 days. Histological characterization revealed heterogenous GAG matrix deposition in *macro*-pellet sections, suggesting that BMSC differentiation occurred at different rates, depending on the spatial location of cells within the *macro*-pellet. A range of metabolite and signal gradients across the diameter of *macro*-pellets likely contributed to this heterogeneity. Cartilage-like matrix formed first at the outer edge or in small localized regions of *macro*-pellets, and incrementally throughout the *macro*-pellets if TGF-β1 exposure was extended. Heterogeneous matrix accumulation observed in *macro*-pellets in our study is similar to previous reports[4,5,15], and is the common rationalization for the use of multi-week cultures with continuous TGF-β exposure. By contrast, when BMSCs were induced as *micro*-pellets, the cells were more responsive to TGF-β1. A single day of TGF-β1 exposure resulted in *micro*-pellets having relatively uniform cartilage-like matrix distribution rich in GAG, with mature lacunae structures, similar to *micro*-pellets exposed to TGF-β1 for 21 days.

The response of ACh to TGF-β1 exposure in *micro*- vs. *macro*-pellet culture was not as profound as that of BMSCs. A single day of TGF-β1 exposure was insufficient to maximize the cartilage-like matrix output. Instead, both ACh *micro*- and *macro*-pellets benefited from extended TGF-β1 exposure, increasing both DNA and GAG content. Because ACh are a highly committed cell type[34], unlike multipotent BMSCs, TGF-β1 serves more to stimulate matrix production rather than guide cell fate. It is possible that lower passage ACh may have a more profound response to short-term TGF-β1 exposure, compared with high passage ACh, used in our study. However, the number of ACh that can be obtained from healthy individuals is a general limitation, and part of the reason why BMSCs are being actively investigated as an alternative starting source for regenerating cartilage tissue.

We evaluated chondrogenic and hypertrophic gene expression from in vitro tissues using qPCR and, following in vivo incubation in NSG mice, tissues were characterized using histology and microCT. Brief, 1–3-day exposure of BMSC *micro*-pellets to TGF-β1 was sufficient to upregulate both chondrogenic (*SOX9, COL2A1*, and *ACAN*) and hypertrophic (*COL10A1*) gene

expression to levels seen in cultures exposed to TGF-β1 for the full 21 days of induction. To confirm that brief TGF-β1 exposure induced hypertrophic tissue formation in BMSCs, we incubated BMSCs and ACh *micro*-pellets subcutaneously in NSG mice. BMSC *micro*-pellets, but not ACh *micro*-pellets, formed hypertrophic bone-like tissue in vivo, regardless of TGF-β1 exposure time. Bone tissue exhibited mineralization and supported characteristic small pockets of marrow, as previously described[20,42]. The increase in hypertrophic gene expression in BMSC exposed to 1-to-3 days of TGF-β1 in vitro was consistent with the presence of mineralized tissue subsequently observed in vivo.

To better understand the transcriptional response to differential TGF-β1 exposure, we performed RNA-seq on BMSCs and ACh *micro*-pellets. We analyzed data from cultures prior to induction, continuously exposed to TGF-β1 and harvested at day 1, 3, 7, or 21, as well as tissues cultured until day 21 where TGF-β1 had been withdrawn at day 1, 3, or 7. This design provided insight into the temporal progression of the differentiation process, as well as opportunity to assess manifestation of programming triggered by different durations of TGF-β1 exposure. Each of the three unique BMSC donors generated a similar gene expression signature, progressively upregulating chondrogenic and hypertrophic genes with 1, 3, 7, or 21 days of TGF-β1 exposure. We assessed gene expression related to cartilage and bone development that was significantly differentially expressed more than 2 log-fold between BMSCs and ACh following TGF-β1 exposure and categorized them by their function including those related to transcription factors, soluble signals and receptors, and ECM molecules and ECM biosynthesis.

*SP7*, or Osterix, is a transcription factor that drives hypertrophy[43] and is required for bone formation[44], and was found to be exclusively upregulated in BMSCs in our study, with as little as 1 day of TGF-β1 exposure. One day of TGF-β1 exposure resulted in a 9.7 log-fold difference in *SP7* expression in BMSCs, compared with ACh cultures, and remained highly expressed following TGF-β1 withdrawal from BMSCs. Because *SP7* was only highly expressed following TGF-β1 induction and maintained this level of expression even after TGF-β1 withdrawal, SP7 expression might be the most reliable early transcription factor to detect BMSC hypertrophic fate. *RUNX2*, the master regulator of osteoblast differentiation, was expressed in our BMSC cultures 6.4 log-fold more than in ACh cultures prior to chondrogenic induction. This baseline expression of *RUNX2* did not increase significantly in BMSCs following TGF-β1 exposure, suggesting that RUNX2 does not induce hypertrophy in BMSCs on its own, but likely requires TGF-β1 mediated upregulation of other transcription factors, such as SP7, for hypertrophic conversion. Previous studies have shown that knockdown of *Sp7* in mice results in diminished hypertrophy in bone progenitor cells in vivo[43]. While molecular inhibitors of SP7 have not been reported, it would be of interest to determine whether regulation of SP7 through knockdown in BMSCs could lead to high-quality, stable hyaline cartilage tissue that does not undergo hypertrophy.

We also identified the expression of soluble signaling factors associated with hypertrophy in BMSCs belonging to the WNT and TGF-β1 superfamily signaling pathways, including *BMP2* and *WNT11*. Inhibition of these factors in BMSC chondrogenic induction cultures has been previously reported with some success[15,40]. Further improvement may be realized using combinations of these inhibitors in *micro*-pellet cultures as described here, at the time of pellet initiation, as hypertrophic fate commitment occurs with a single day of TGF-β1 exposure in BMSCs.

ECM molecules associated with BMSC hypertrophy and mineralization, such as *COL10A1* and *IBSP*, were not statistically differentially expressed between BMSC and ACh until more than 1 day of TGF-β1 exposure. ECM molecules and their biosynthesis

are generally downstream products of transcriptional and receptor signaling machinery and, as such, it is not surprising that a relative delay in their gene expression might be observed. Expression of ECM associated genes can serve as good markers for assessing cartilage hypertrophy and tissue quality, but upstream targets are more likely, and feasible, to successfully mitigate hypertrophy.

Comparative analysis of RNA-seq data from BMSC and ACh *micro*-pellets provides a number of useful insights into the pathways that differ between these cell population. It identifies potential target pathways to obstruct BMSC hypertrophy, and genes that could be used in reporter assays to facilitate the development of chondrogenic media, cell processing, and scaffolding.

While BMSCs have significant unrealized potential in cartilage tissue engineering, current chondrogenic differentiation protocols yield sub-optimal cartilage-like tissue with a hypertrophic propensity[1]. Using a *micro*-pellet model, we show that BMSC chondrogenic kinetics are significantly more rapid than historical *macro*-pellet data suggests, and that BMSC chondrogenic and hypertrophic commitment is instructed by a single day of TGF-β1 exposure. This highly relevant study demonstrates that: (1) *macro*-pellets, which are large heterogeneous tissue models, confound the differentiation kinetics of BMSCs that are visible in *micro*-pellet models; (2) even a single day of TGF-β1 exposure drives BMSC to form hypertrophic tissue in vivo, requiring early intervention to prevent hypertrophy; and (3) ACh and BMSCs respond distinctly to TGF-β1. Future efforts to generate stable cartilage-like tissue using BMSCs should use small diameter, homogeneous models, and focus on manipulation of the culture conditions over the first few hours of culture induction, including early efforts to obstruct hypertrophy. Our data highlight logical genetic or pathway targets that may facilitate generation of stable cartilage-like tissue from BMSC. Alternative strategies that utilize different signaling pathways either in conjunction with TGF-β1 or independent of TGF-β1 need to be considered for cartilage tissue regeneration.

## Methods

**Microwell-mesh fabrication**. The Microwell-mesh culture plates were fabricated as described previously[5]. The Microwell-mesh inserts used in this study had arrays of pyramidal microwells ($2 \times 2$ mm square $\times 0.8$ mm deep), covered with a nylon mesh. The microwells enable the simultaneous and efficient formation of many cellular *micro*-pellets, while the nylon mesh prevents displacement of the *micro*-pellets from microwells over long-term culture (see Supplementary Movie 1). In brief, fabrication and sterilization utilized methods as follows. A thin 3 mm layer of polydimethylsiloxane (PDMS, Sylgard184; Dow Corning) was cast on a polystyrene microwell negative template and cured for 60 min at 80 °C. Discs (35 mm) were punched from PDMS sheets. Nylon mesh (nylon 6/6, part number: CMN-0035 Amazon.com) having 36 μm square openings was bonded over the microwells with silicone glue (Selleys, Aquarium Safe). Excess mesh was trimmed from the edge of the PDMS discs, and they were anchored into 6-well plates (Corning) with a dab of silicone glue. Plates were sterilized in 70% ethanol and washed thoroughly with PBS. Prior to seeding cells, wells with inserts were incubated in a sterile solution of 5% Pluronic F-127 (Sigma-Aldrich) in PBS to prevent cell attachment to the PDMS surface[45], thus promoting cell aggregation.

**Human BMSC isolation and expansion**. BMSC cultures were established using 20 mL of heparinized bone marrow aspirate (BMA) collected from the iliac crests of consenting volunteer healthy adult human donors at the Mater Hospital, Brisbane, Australia. Ethics approval for aspirate collection was granted by the Mater Health Services Human Research Ethics Committee and the Queensland University of Technology Human Ethics Committee (Ethics number: 1000000938), as per the Australian National Health and Medical Research Council guidelines. BMSCs were collected as previously described[5], with the exception of donor 3, which was not enriched for mononuclear cells prior to plastic attachment overnight. BMSC donor details were as follows: 24-year-old male (BMSC 1), 44-year-old male (BMSC 2), 21-year-old female (BMSC 3), and 43-year-old male (BMSC 4). BMSC expansion medium contained low glucose DMEM, supplemented with GlutaMAX and pyruvate, 10% fetal bovine serum (FBS), 100 U/mL penicillin/streptomycin (PenStrep), all from Thermo Fisher Scientific, 10 ng/mL fibroblast growth factor-1 (FGF-1; Peprotech), and 5 μg/mL porcine heparin sodium salt (Sigma-Aldrich). Cells were

distributed into five T175 culture flasks with 35 mL of expansion medium per flask and were allowed to attach to the plastic surface overnight in a 20% $O_2$, 5% $CO_2$, and 37 °C incubator. The medium was replaced after 24 h to remove loose cells and cell expansion was continued in a reduced oxygen atmosphere, 2% $O_2$, 5% $CO_2$, and 37 °C incubator. At 80% confluence, cells were passaged using 0.25% trypsin/EDTA (Thermo Fisher Scientific) and fresh flasks were re-seeded at ~1500 cells/$cm^2$. All BMSC donor cells had undergone cryopreservation in 90% FBS and 10% DMSO, thawed, and induced at passage 3.

**Human ACh expansion.** Normal human ACh from the knee were purchased from Lonza. Based on the manufacturer's information sheet, the cells were cryopreserved at passage 2, and donor information was as follows: 34-year-old male (Lot#: BF3339; ACh 1 in this study) and 50-year-old male (Lot#: BF3307; ACh 2). After thawing, these cells were grown as described for BMSCs above, in a 2% $O_2$, 5% $CO_2$, and 37 °C incubator and induced at passage 6.

**Chondrogenic-induction cultures.** Chondrogenic induction media consisted of high glucose DMEM supplemented with GlutaMAX and pyruvate, 1× insulin-transferrin-selenium-ethanolamine (ITS-X), PenStrep, all from Thermo Fisher Scientific, 200 μM ascorbic acid 2-phosphate (Sigma-Aldrich), 40 μg/mL l-proline (Sigma-Aldrich), and 10 ng/mL TGF-β1 (Peprotech) where specified. The cells in the "day 0" TGF-β1 removal condition were never supplemented with TGF-β1. The cells were force-aggregated in the well plates by centrifugation at $500 \times g$ for 3 min. Full medium exchanges were performed every 2 days or on specified day of TGF-β1 removal. For TGF-β1 removal, the cell pellets were rinsed once with PBS, then fresh induction medium without TGF-β1 was added. For *macro*-pellet cultures, cells were seeded in 96-well plates containing deep v-bottom wells (Sigma-Aldrich), such that each well or aggregate contained $2 \times 10^5$ cells per well in 1 mL of culture medium. In *micro*-pellet cultures, cells were seeded at $1.25 \times 10^6$ per well of a 6-well plate. At this seeding density, ~250 *micro*-pellets are formed in each well, with each *micro*-pellet containing ~5000 BMSCs. Each well contained 4 mL of total culture medium.

**GAG and DNA quantification.** GAG and DNA was analyzed as previously described[5]. Briefly, tissues were papain (Sigma-Aldrich) digested overnight at 60 °C. GAG was quantified using the 1,9-dimethymethylene blue (Sigma-Aldrich) assay, using chondroitin sulfate from shark cartilage (Sigma-Aldrich) to generate a standard curve. DNA was quantified using the PicoGreen assay kit (Thermo Fisher Scientific). Four replicate samples were analyzed for each unique cell donor and the means were compared to respective day 21 +TGF-β1 control cultures.

**RNA isolation.** Samples were crushed in RLT buffer (Qiagen) containing β-mercaptoethanol (Sigma-Aldrich) and RNA was isolated using the RNeasy Mini Kit (Qiagen) with on-column DNase I (Qiagen) digestion, as per manufacturer's instructions. RNA concentrations were determined using a NanoDrop 1000 (Thermo Fisher Scientific).

**Quantitative-PCR analysis.** cDNA was synthesized from total RNA using SuperScript III First-Strand Synthesis System for RT-PCR (Thermo Fisher Scientific), as per manufacturer's instructions. Quantitative PCR (qPCR) reactions were prepared using SYBR Green PCR Master Mix (Applied Biosystems). Three technical replicates were loaded for each of the four culture well replicates, per donor, in 384-well plates and analyzed on a Viia7 Real Time PCR System (Applied Biosystems). The forward and reverse primer sequence for target genes, as well as the qPCR run parameters have been previously published[5]. Target gene expression was normalized to GAPDH expression and calculated using the following formula, $2^{-(Ct(\text{Gene of interest}) - Ct(GAPDH))}$.

**Subcutaneous implantation of cartilage defect model.** NOD-scid IL2R gamma[null] (NSG) mice[18] were purchased from the Jackson Laboratory and bred in the Animal Facility at the Translational Research Institute (TRI) in Brisbane. The University of Queensland (UQ) and the Queensland University of Technology (QUT) Animal Ethics Committees authorized the animal procedures, as per the Australian National Health and Medical Research Council guidelines. All animal procedures were carried out in accordance with the approved guidelines (Ethics numbers: AEMAR53777 and AEMAR53765). Female mice 6–12 weeks old were used in these studies. Artificial cartilage defect models were prepared from bovine tissue as previously described[5]. Briefly, plugs containing cartilage and bone were drilled out from bovine knees using a 10-mm coring bit. Full thickness cartilage defects were drilled out of the plug using a 3.5-mm drill bit. The defects were washed of debris and sterilized in 70% ethanol for 24 h and washed 3× in PBS over another 24 h. The PBS was discarded, and the defects were kept frozen at −20 °C until use. The defects were filled with 21-day cultured *micro*-pellets that were grown in the presence of TGF-β1 for 21 days or had TGF-β1 withdrawn at earlier time-points, and sealed with fibrin glue (one drop of fibrinogen, followed by one drop of thrombin) (Tisseel, Baxter). Each defect was implanted subcutaneously in a pocket made on the back of an anaesthetized NSG mouse and the skin stapled to close the wound. These tissues were permitted to incubate in NSG mice for 8 weeks, at which point the animals were euthanized and the tissues recovered for analysis.

**MicroCT analysis.** Tissues excised from mice were fixed in 4% paraformaldehyde (PFA) for 24 h and scanned in plastic tubes containing 70% alcohol and styrofoam to reduce movement. MicroCT analysis was performed to detect hard tissue formation using an Inveon PET-CT Scanner (Siemens; 27.6 μm pixel, 60 kV, 350 μA, 2.5 s exposure) or the SkyScan 1272 (Bruker; 17-22 μm pixel, 50 kV, 200 μA, 0.25 mm Al filter, 425 ms exposure). Reconstruction and imaging were performed using Inveon Software (Siemens) or NRecon Software and DataViewer Sofware (Skyscan Bruker), for related machines.

**Histology.** Tissues induced for 21 days were fixed in 4% PFA for 30 min, washed and frozen in Tissue-Tek OCT compound (Sakura Finetek). Samples were cryosectioned at 7 μm and collected onto poly-lysine coated slides (Thermo Fisher Scientific). After microCT analysis, tissues excised from mice were decalcified in EDTA solution until soft, dehydrated, paraffin embedded and sectioned at 5 μm. Sections were stained with Toluidine blue (Sigma) to detect GAG.

**RNA-seq.** Total RNA was collected from BMSC and ACh cultures prepared in Microwell-mesh plates on days 0, 1, 3, 7, and 21, with or without TGF-β1 washout, as indicated. "Day 0" cells represent expanded cells that were not induced. RNA integrity was confirmed using the Agilent 2100 Bioanalyser (Agilent Technologies). Next-generation sequencing and bioinformatics was performed by the Australian Genome Research Facility (AGRF, Melbourne) using the Illumina HiSeq 2500 platform (100 bp single end run). AGRF analysis involved demultiplexing, quality control, and then data was processed through RNA-seq expression analysis workflow, which included alignment, transcript assembly, quantification, normalization, and differential gene expression analysis (Bioconductor R package edgeR[46]). Significance of differentially expressed genes between ACh and BMSC cultures was considered at an adjusted P-value (FDR) of <0.05. Heatmaps were generated using the heatmap.2 function in the R gplots package[47]. Raw data are deposited in the Gene Expression Omnibus (GEO), accession number GSE161176.

**Statistics and reproducibility.** Where data are presented as box plots, whiskers represent the minimum and maximum value, the center represents the median, and the edges of the box represent the first and third quartiles ($n = 4$ for each cell donor). Replicate donor information is provided in each figure caption. Statistical analysis was performed using ordinary one-way ANOVA with Dunnett's multiple comparisons test in GraphPad Prism version 7. P-values of 0.05 or less were considered statistically significant.

**Human ethics approval and consent to participate.** Bone marrow aspirates were collected from informed and consenting volunteer healthy adult human donors at the Mater Hospital, Brisbane, Australia. Ethics approval for aspirate collection was granted by the Mater Health Services Human Research Ethics Committee and the Queensland University of Technology Human Ethics Committee (Ethics number: 1000000938), as per the Australian National Health and Medical Research Council guidelines. ACh were purchased from Lonza.

**Animal ethics.** NSG mice were purchased from the Jackson Laboratory and bred in the Animal Facility at the Translational Research Institute (TRI) in Brisbane. The University of Queensland (UQ) and the Queensland University of Technology (QUT) Animal Ethics Committees reviewed and approved the animal ethics applications, as per the Australian National Health and Medical Research Council guidelines. All animal procedures were carried out in accordance with the approved guidelines (Ethics numbers: AEMAR53777 and AEMAR53765).

**Reporting summary.** Further information on research design is available in the Nature Research Reporting Summary linked to this article.

## Data availability

Data supporting the conclusions of this paper are available from the corresponding author upon request. Raw RNA-seq data have been uploaded to NCBI database, and can be found at accession number: GSE161176. Data comparing the differentially expressed genes between BMSC and ACh are provided in the tables contained in Supplementary Data 1.

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

## Acknowledgements

The Translational Research Institute (TRI) is supported by Therapeutic Innovation Australia (TIA). TIA is supported by the Australian Government through the National Collaborative Research Infrastructure Strategy (NCRIS) program. K.F. and M.R.D. thank the TRI Biological Resource Facility for help with animal studies, the TRI Preclinical Imaging Facility for help with microCT analysis, the TRI Histology Facility for help with tissue processing and sectioning, the Mater Hospital for BMA collection, and the Australian Genome Research Facility (AGRF) for RNA-seq and bioinformatics analysis. K.F., M.R.D., R.W.C., and T.J.K. gratefully acknowledge project support from the National Health and Medicine Research Council (NHMRC) of Australia (Project Grant APP1083857) and NHMRC Fellowship support of M.R.D. (APP1130013). K.F. and P.G.R. are supported by the Division of Intramural Research (DIR) of the National Institute of Dental and Craniofacial Research (NIDCR), a part of the Intramural Research Program (IRP) of the National Institutes of Health (NIH), Department of Health and Human Services (DHHS) (1 ZIA DE000380 35). We thank medical-animations.com for generously producing Supplementary Movie 1 with guidance from Ms. Ena Music.

## Author contributions

K.F., P.G.R., T.J.K., R.W.C., and M.R.D. designed the research, analyzed the data, and wrote the paper; K.F. and M.R.D. performed the research.

## Competing interests

The authors declare no competing interests.
