## [Peer Review File · Communications Biology]

Reviewers' comments:

Reviewer #1 (Remarks to the Author):

Overview: In this study Futrega and colleagues challenge the long established chondrogenic differentiation protocol used to induce MSC differentiation. In this study they show that micro-pellets of MSCs can be efficiently induced to form cartilage with just 24 hours of exposure to TGF-B1. This stands in stark contrast with the classic protocols that call for 2-3 weeks of continuous TGF-B1 exposure and rely on larger cell pellets. By reducing the size of the micropellet, diffusion gradients were substantially reduced resulting in more homogenous timing of differentiation induction across the entire micro-tissue. Overall, this is an important piece of work that challenges a long standing protocol that has not been critically evaluated, until now. This work and the analysis provide compelling evidence in favor of rethinking the way we induce BM-MSC chondrogenesis and also shows that there remains a significant gap between the chondrocyte derived cartilage and BM-MSC derived cartilage in terms of both matrix composition and gene expression. I have a number of relatively minor concerns that need to be corrected to prevent misinterpretation of the work.

Major Concerns

Fig 4. The caption states the values are Mean \pm SD but the plots appear to be box and whisker plots. In addition, the caption says the values represent delta-CT compared to GAPDH (Ct(gene of interest)- Ct(GAPDH)). As GAPDH is normally abundant and has a very low CT, the smaller delta CT would be interpreted as higher abundance and the higher delta CT would represent the gene has a lower expression. Your discussion however describes high delta CT to indicate high gene expression, which makes me think your description of your PCR quantification is incomplete. Can you please clarify so the direction of gene expression changes is clear for the reader.

Minor Concerns

1. Stats in 2A don't appear to be necessary and add clutter to an already compact graph. Unless you had predefined hypothesis about which duration of exposure would generate differences in size, I suggest removing the stats and allowing the data to stand on its own.
2. For 2C and 3C, The large # of **** present in the figure are quite distracting. I suggest using just 1 symbol, whatever you selected as your alpha in your experimental planning. Whether your P-value is much smaller than the alpha or just slightly smaller than your alpha doesn't change statistical interpretation as the result is either significant or not significant. Your methods describe just 1 alpha value, so a single symbol that it was below or above that symbol should suffice.
3. Red arrows are not visible in Fig. 3A.
4. The results section reads like it wants to compare BM-MSC to ACh cultures, yet the groups are never directly compared in a figure. You might consider directly comparing these groups to better illustrate the comparison. Right now, the reader needs to bounce between Fig 2 and Fig 3 to see the differences the authors are highlighting in the text.
5. Figure 5. 'Micro-CT analysis indicating a tissue density similar to that of the bovine bone in the artificial cartilage defect model (Fig. 5a). While not mineralized, GAG staining of ACh micro - pellets was faint (Fig. 5b),' Panel 5b is the micro-CT while 5a is the GAG staining. Either the text of figure needs to be flipped so they are in agreement.
6. Fig 6B, it is very difficult to see the 'open blue circle's much of the font and data points in this manuscript are very small and require extensive magnification to interpret.

Reviewer #2 (Remarks to the Author):

In this manuscript, the author used the Microwell-mesh, to establish small diameter micro-pellets to study the differentiation of BMSCs into chondrocytes. BMSC macro-pellet and micro-pellet cultures were exposed to TGFb for different time points and compared after 21 days for traditional in vitro and in vivo chondrogenic parameters. This is a very well designed study with clear stepwise identification of in vitro chondrocyte formation, however, the authors did not show obvious high quality cartilage-like tissue from BMSC with 1 day exposure with TGFb1, and the study is rather a comparative study between different exposure times of TGFb1. Thus, the title and the conclusion of the manuscript should be modified to show the impact of the treatment of BMSC with TGFb1 at different exposure times on chondrogenesis. The in vivo part of this study should be supported by more immune-staining for both bone and cartilage specific matrix proteins. The authors should discuss their in vivo data in the discussion section in more details and within the scope of how their data can be beneficiary in the application of BMSCs in clinical cartilage repair.

Reviewer #3 (Remarks to the Author):

Dr. Doran and colleagues make a strong case for the chondrogenic and hypertrophic commitment of bone marrow stromal cells (BMSCs) as early as 1 day after TGFb exposure. They justify the use of a smaller-sized pellet assay, termed a micro-pellet, to parse out early time points as indication of differentiation driven by TGFb. In this sense, this shows a technical advance that could mitigate excessive cost, which would be useful for smaller labs where cost is a significant concern. A significant strength of the paper to note is their RNA-sequencing data (Figure 6), which supports their hypothesis that BMSCs turn on and maintain fate-specific genes as early as 1 day after TGFb exposure, compared to 21 days of TGFb exposure. However, the paper's claims do not inherently advance the current conceptual knowledge or advantage within the field. For instance, the first sentence of the abstract clearly identifies a deficit in the current space, yet the hypothesis and work do not openly address this point.

Intuitively, it is reasonable that a reduced surface area from fewer cells has more propensity to be influenced by exogenous TGFb signaling, thus influencing responsiveness and cell commitment (Fig. 1 & 2). However, would cells primed with TGFb for 1 day in 2D, then pelleted, form an equivalent product? By making pellets of varying size, one could demonstrate whether it is the availability of TGFb, as the authors claim, that determine homogeneity and overall pellet chondrogenic capability, or the macro- vs micro- size of the pellet that is the true determinant. Furthermore, one would have to explore the mechanism of why TGFb is required for only 1 day to truly influence the current field with their hypotheses, as well as addressing why other factors, such as BMP2, was omitted from the study.

Another notable point to consider addressing would be the authors' choice to use such highly passaged articular chondrocytes. It is well known that articular chondrocytes are fully differentiated, but it would be a more direct comparison with BM-MSCs if lower passages (P0-1) were used, which retains more biologically relevant profiles. If the relevant assays repeated with low passage articular chondrocytes resulted in the same trends as the authors show, this would strengthen their hypothesis. Although conceded on page 12, line 9, more in-depth justification in this section or the discussion would help defend the authors' methodology.

Inclusion of 1 day of TGFb exposure in Figure 5 would help support the overall hypothesis that it is sufficient to drive differentiation, since the figure begins at 3 days of TGFb exposure. IHC/IFC characterization of tissue formed (i.e. perilipin, alkaline phosphatase, osterix, etc.) in Figure 5 would also be beneficial to definitively observe mineralization detected in the Micro-CT scan, thus supporting their hypothesis that minimal exposure of TGFb is sufficient to drive hypertrophic tissue formation.

In regards to Figure 6, it would be useful if, instead of describing all theoretically possible factors for hypertrophy regulation, a list of key possible factors that drive hypertrophic mechanisms would

be advantageous. With this depicted summary, relevant blockers (WNT and/or BMP blockers) used in vitro could validate the authors' claims of value for identifying these key factors, to ensure the quality of cartilage that one would need for clinical significance.

As far as technicalities, Page 7 line 27-28 specifies the use of red arrows in Figure 3a, however there are no red arrows included in the image. Figure 4 seems to be a duplicate panel. On page 18, line 5, it is assumed that the authors meant "Briefly," rather than "Brief,". On page 21, line 13, it is also assumed that a Greek letter B will be used in lieu of "TGF-B1". In general, the authors may want to consider shortening/condensing the Discussion section; it may be more impactful if the overall messages were discussed and presented in a more concise manner.

Editor summary

In summary, all referees agree in that the manuscript poses a technological advance for the field. However, they raise important concerns that we believe should be addressed. In particular, reviewer #2 and #3 state that the authors should support the one-day timepoint with additional data. In addition, reviewer #2 requests further characterization of the *in vivo* findings and reviewer #3 requests two experiments that would strengthen the conclusions by controlling for the effect of pellet size and the number of passages. Lastly, reviewer #1 noticed some important textual clarifications that should be addressed.

Referee expertise:

Referee #1: mesenchymal stem cell, bioengineering, cell biology

Referee #2: stem cells and bone marrow-derived mesenchymal stem protocols

Referee #3: bone regeneration, chondrocytes, stem cell biology

General comment: We are grateful for the detailed review. Our response to Referee comments or questions is in blue text, and new manuscript text is highlighted in yellow. Please note, that in response to the Referees' suggestions we have made minor edits to Figures 2, 3, and 4. We have added new data to Figure 5. These data show that BMSC exposed to a single day of TGF- β 1 during *in vitro* culture form mineralised tissue when implanted *in vivo*.

Referee #1 (Remarks to the Author):

Overview: In this study Futrega and colleagues challenge the long established chondrogenic differentiation protocol used to induce MSC differentiation. In this study they show that micro-pellets of MSCs can be efficiently induced to form cartilage with just 24 hours of exposure to TGF-B1. This stands in stark contrast with the classic protocols that call for 2-3 weeks of continuous TGF-B1 exposure and rely on larger cell pellets. By reducing the size of the micropellet, diffusion gradients were substantially reduced resulting in more homogenous timing of differentiation induction across the entire micro-tissue. Overall, this is an important piece of work that challenges a long standing protocol that has not been critically evaluated, until now. This work and the analysis provide compelling evidence in favor of rethinking the way we induce BM-MSc chondrogenesis and also shows that there remains a significant gap between the chondrocyte derived cartilage and BM-MSc derived cartilage in terms of both matrix composition and gene expression. I have a number of relatively minor concerns that need to be corrected to prevent misinterpretation of the work.

Major Concerns

Referee comment: Fig 4. The caption states the values are Mean \pm SD but the plots appear to be box and whisker plots.

Author response: Sorry, we converted the graphs to box plots, but overlooked updating this detail in the caption. We have removed the statement about Mean \pm SD and replaced with "Box plot" in the caption.

Referee comment: In addition, the caption says the values represent delta-CT compared to GAPDH (Ct(gene of interest)- Ct(GAPDH)). As GAPDH is normally abundant and has a very low CT, the smaller delta CT would be interpreted as higher abundance and the higher delta CT would represent the gene has a lower expression. Your discussion however describes high delta CT to indicate high gene expression, which makes me think your description of your PCR quantification is incomplete. Can you please clarify so the direction of gene expression changes is clear for the reader.

Author response: Thank you for catching this. Values are generated using the "delta Ct method". The formula we used to derive the plotted gene expression values is: $2^{-(\text{Ct}(\text{Gene of interest})-\text{Ct}(\text{GAPDH}))}$ and therefore

the values represent $2^{\Delta\Delta Ct}$. We apologise for the confusion and have updated this in the caption of Figure 4 and added the formula in the qPCR methods section.

Referee comment:

1. Stats in 2A don't appear to be necessary and add clutter to an already compact graph. Unless you had predefined hypothesis about which duration of exposure would generate differences in size, I suggest removing the stats and allowing the data to stand on its own.

2. For 2C and 3C, The large # of **** present in the figure are quite distracting. I suggest using just 1 symbol, whatever you selected as your alpha in your experimental planning. Whether your P-value is much smaller than the alpha or just slightly smaller than your alpha doesn't change statistical interpretation as the result is either significant or not significant. Your methods describe just 1 alpha value, so a single symbol that it was below or above that symbol should suffice.

Author response: We have reduced the number of asterisks to one in Figures 2, 3, and 4. Thank you for the suggestion.

Referee comment: Red arrows are not visible in Fig. 3A.

Author response: This has been modified to make the red arrows visible.

Referee comment: The results section reads like it wants to compare BM-MSc to ACh cultures, yet the groups are never directly compared in a figure. You might consider directly comparing these groups to better illustrate the comparison. Right now, the reader needs to bounce between Fig 2 and Fig 3 to see the differences the authors are highlighting in the text.

Author response: It was not our intent to make the focus of this manuscript BMSC versus ACh. ACh data is provided as a reference point. In our results, we described the BMSC (Figure 2) data in detail (first paragraph on page 7). We then briefly described ACh (Figure 3) in the second paragraph and point out some differences in patterns, relative to BMSCs. Instead of plotting the BMSC and ACh data side-by-side, we put multiple replicate BMSC data sets in the same figure. Our intent was to convey this highly reproducible response of the BMSCs to different durations of TGF- β 1 exposure, and that it is qualitatively different from that of ACh.

Referee comment: Figure 5. 'Micro-CT analysis indicating a tissue density similar to that of the bovine bone in the artificial cartilage defect model (Fig. 5a). While not mineralized, GAG staining of ACh micro-pellets was faint (Fig. 5b),' Panel 5b is the micro-CT while 5a is the GAG staining. Either the text of figure needs to be flipped so they are in agreement.

Author response: We've corrected the text in the results. Thank you.

Referee comment: Fig 6B, it is very difficult to see the 'open blue circle's much of the font and data points in this manuscript are very small and require extensive magnification to interpret.

Author response: Figure 6B has been enlarged, and the relative size of Figure 6B font and data points have also been enlarged.

Referee #2 (Remarks to the Author):

In this manuscript, the author used the Microwell-mesh, to establish small diameter micro-pellets to study the differentiation of BMSCs into chondrocytes. BMSC macro-pellet and micro-pellet cultures were exposed to TGF β for different time points and compared after 21 days for traditional in vitro and in vivo chondrogenic

parameters. This is a very well designed study with clear stepwise identification of *in vitro* chondrocyte formation, however, the authors did not show obvious high quality cartilage-like tissue from BMSC with 1 day exposure with TGF β 1, and the study is rather a comparative study between different exposure times of TGF β 1.

Referee comment: Thus, the title and the conclusion of the manuscript should be modified to show the impact of the treatment of BMSC with TGF β 1 at different exposure times on chondrogenesis.

Author response: The key message in our paper is that BMSC differentiation is induced after just a single day of TGF- β 1 exposure in chondrogenic cultures. Further exposure to TGF- β 1, for varying lengths of time, does not help maintain the stability of cartilage-like tissue, and by day 21, all exposure times trialled yielded similar tissues. Our title was selected in an effort to communicate this within the character limit restrictions. We have considered other titles, but believe that the current title (which is within the character limit) will be the most effective in attracting readers to this paper.

Referee comment: The *in vivo* part of this study should be supported by more immune-staining for both bone and cartilage specific matrix proteins.

Author response: We have added the additional day 1 time point data to Figure 5. These critical *in vivo* data show that histologically identifiable bone and bone marrow tissue are formed and that *micro*-pellets mineralized. Mineralization is the most important feature of cartilage hypertrophy, and these data demonstrate that a single day of TGF- β 1 exposure triggers this outcome in BMSCs. Assessing the quality of the newly formed bone is beyond the scope of this study. If high-quality bone were the objective, other bone tissue engineering strategies would likely be superior.

Referee comment: The authors should discuss their *in vivo* data in the discussion section in more details and within the scope of how their data can be beneficiary in the application of BMSCs in clinical cartilage repair.

Author response: A key message of the paper is that BMSCs stimulated with TGF- β 1 yield hypertrophic tissue; and we do not recommend this method of BMSC chondrogenic induction as a strategy for generating tissue for clinical cartilage repair. Our data, and discussion, highlight logical genetic or pathway targets that may facilitate generation of stable cartilage-like tissue from BMSC.

Our *in vivo* data (histology and microCT) show that the microtissues formed hypertrophic and mineralized tissue. We have modified the Discussion paragraph as highlighted in yellow below (page 7, line 8):

We evaluated chondrogenesis and hypertrophy in *in vitro* tissues using qPCR and following *in vivo* incubation in NSG mice using histology and microCT analysis. Brief, 1-to-3 day exposure of BMSC *micro*-pellets to TGF- β 1 was sufficient to upregulate both chondrogenic (*SOX9*, *COL2A1*, and *ACAN*), and hypertrophic (*COL10A1*) gene expression to levels seen in cultures exposed to TGF- β 1 for the full 21 days of induction. To confirm that brief TGF- β 1 exposure induced hypertrophic tissue formation in BMSC, we incubated BMSC and ACh *micro*-pellets subcutaneously in NSG mice. BMSC *micro*-pellets, but not ACh *micro*-pellets, formed hypertrophic bone-like tissue *in vivo*, regardless of TGF- β 1 exposure time. Bone tissue exhibited mineralization and supported characteristic small pockets of marrow, as previously described^{37,38}. The increase in hypertrophic gene expression in BMSC exposed to 1-to-3 days of TGF- β 1 *in vitro* was consistent with the subsequent formation of mineralized tissue *in vivo*.

We have also added a final sentence to the Discussion (page 19, line 11):

While BMSCs have significant unrealized potential in cartilage tissue engineering, current chondrogenic differentiation protocols yield sub-optimal cartilage-like tissue with a hypertrophic propensity [1]. Using a *micro*-pellet model, we show that BMSC chondrogenic kinetics are significantly more rapid than historical *macro*-pellet data suggests, and that BMSC chondrogenic and hypertrophic commitment is instructed by a single day of TGF- β 1 exposure. This highly relevant study demonstrates that: (1) *macro*-pellets, which are large heterogeneous tissue models, confound the differentiation kinetics of BMSCs that are visible in *micro*-

pellet models; (2) even a single day of TGF- β 1 exposure drives BMSC to form hypertrophic tissue *in vivo*, requiring early intervention to prevent hypertrophy; and (3) articular chondrocytes and BMSCs respond distinctly to TGF- β 1. Future efforts to generate stable cartilage-like tissue using BMSCs should use small diameter, homogeneous models, and focus on manipulation of the culture conditions over the first few hours of culture induction, including early efforts to obstruct hypertrophy. Our data highlight logical genetic or pathway targets that may facilitate generation of stable cartilage-like tissue from BMSC. Strategies that exploit different signally pathways, either in conjunction with TGF- β 1 or independent of TGF- β 1, will be required to form cartilage from BMSC.

Referee #3 (Remarks to the Author):

Dr. Doran and colleagues make a strong case for the chondrogenic and hypertrophic commitment of bone marrow stromal cells (BMSCs) as early as 1 day after TGF β exposure. They justify the use of a smaller-sized pellet assay, termed a micro-pellet, to parse out early time points as indication of differentiation driven by TGF β . In this sense, this shows a technical advance that could mitigate excessive cost, which would be useful for smaller labs where cost is a significant concern. A significant strength of the paper to note is their RNA-sequencing data (Figure 6), which supports their hypothesis that BMSCs turn on and maintain fate-specific genes as early as 1 day after TGF β exposure, compared to 21 days of TGF β exposure.

Referee comment: The paper's claims do not inherently advance the current conceptual knowledge or advantage within the field. For instance, the first sentence of the abstract clearly identifies a deficit in the current space, yet the hypothesis and work do not openly address this point.

Author response: It is true that we do not provide a solution for how to generate stable cartilage from BMSC. However, we do provide two important insights which we hope will guide the field towards solutions.

- 1. Micro-pellet models:** We reason that the routine use of large diameter *macro*-pellets, or large tissue models, has confounded fundamental insights into how BMSCs respond to chondrogenic induction factors. Virtually all studies are fixated on optimising BMSC chondrogenesis over many days, to weeks. Using a *micro*-pellet model, we show that BMSCs respond to brief (1-3 days) exposure to TGF- β 1, and that further TGF- β 1 stimulation is not beneficial. BMSC response to brief growth factor stimulation is obscured in *macro*-pellet models.
- 2. Culture manipulation at early time points:** Data from *macro*-pellet models would motivate researchers to optimise growth factor combinations over several days or weeks of culture, and historical studies have optimised over several weeks. Our data adds value to the literature by identifying that optimisation over the first hours or days of induction is more likely critical, including early obstruction of hypertrophy, and we discuss this in our manuscript.

Referee comment: Intuitively, it is reasonable that a reduced surface area from fewer cells has more propensity to be influenced by exogenous TGF β signalling, thus influencing responsiveness and cell commitment (Fig. 1 & 2). However, would cells primed with TGF β for 1 day in 2D, then pelleted, form an equivalent product?

Author response: Thank you for the question; we also considered a variation on the proposed experiment. In a pilot study, we added TGF- β 1 to a BMSC expansion culture (which consisted of low glucose DMEM and 10% FBS) in a flask at near confluence for 3 days, and then performed *macro*- and *micro*-pellet cultures (in serum-free chondrogenic media). Pellets generated from BMSCs exposed to TGF- β 1 during monolayer culture failed to generate cartilage-like tissue, while control BMSC that had not been exposed to TGF- β 1 during monolayer culture formed cartilage-like tissue, similar to those described in our manuscript. Because this pilot study did not generate positive outcomes, we only captured images (see below) and performed no further analysis. It is possible that this might work if one used chondrogenic induction media instead of serum-containing expansion media to prime the cells. However, a number of other conditions may have to

be optimized in addition to media formulation, such as cell confluence, adding Rho/Rho-associated protein kinase (ROCK) inhibitor or controlling surface tension (see [2]), etc., and we believe that this is outside of the scope of our current study.

Referee comment: By making pellets of varying size, one could demonstrate whether it is the availability of TGF β , as the authors claim, that determine homogeneity and overall pellet chondrogenic capability, or the macro- vs micro- size of the pellet that is the true determinant.

Author response: We do not claim that TGF- β 1 availability is the major or sole limitation in *macro*-pellet cultures.

Page 6, line 136: “Larger diameter macro-pellets inherently suffer from increased diffusion gradients of metabolites, gases, and other factors (Fig. 1c) [3, 4], while gradients are reduced in smaller diameter micro-pellets, yielding more homogeneous cartilage-like tissues [5, 6].”

Macro-pellets self-assemble over the first few hours of culture, becoming so large, that a number of nutrients/metabolites are likely diffusion-limited. We have not compared a series of different pellet sizes side by side. However, if different sized pellets were made, each size would suffer from slightly different diffusion gradients, influencing the supply of a range of metabolites or autocrine signals. Our argument is not that the limitation of specific nutrient or signal cause the observed outcome, but rather that reducing these limitations in a *micro*-pellet culture reveals that only a, brief, single day of TGF- β 1 exposure is required to induce differentiation.

Referee comment: Furthermore, one would have to explore the mechanism of why TGF β is required for only 1 day to truly influence the current field with their hypotheses, as well as addressing why other factors, such as BMP2, was omitted from the study.

Author response: We used TGF- β in our study because it is the most commonly used growth factor for BMSC chondrogenic induction. TGF- β activates a number of known pathways (e.g. Smad signalling [7]) that stimulate chondrogenic/hypertrophic differentiation. We imagine that the mechanism of TGF- β action in our study is the same as previously described [7], whereby a differentiation cascade involving numerous transcriptional and molecular pathways is at play. It is evident from our RNA-seq data that the gene expression profile of induced vs. non-induced (Day 0) BMSCs changes significantly (see Figure 6B, MDS plot). In Figure 6B, we highlight differentially expressed genes (BMSC vs. ACh) that have been previously annotated as being related to chondrogenic/osteogenic differentiation (Figure 6B, heatmap). We believe that the mechanism of chondrogenic induction and hypertrophy is similar in our study and previous studies, it is just that our data reveal that this machinery does not require the classically assumed 14-21 days of TGF- β

stimulation to be manifest in BMSCs. In other words, the fate of the BMSCs is determined after only one day, and is not further altered with longer treatment.

BMP2, or other TGF- β family members, were not purposefully omitted from the study. Others have found similar BMSC differentiation outcomes when comparing TGF- β and BMP2 [8]. It is possible that brief BMP2 exposure could produce similar outcomes, as we have observed with TGF- β in our current study, but we have not tested this. Also, because we were interested in chondrogenesis, and not hypertrophy, we would not have considered using BMP2. BMP2 is thought to be a stimulator of hypertrophy and osteogenesis, and others have suggested that BMP2 should specifically be inhibited to achieve stable cartilage-like tissue [9].

Referee comment: Another notable point to consider addressing would be the authors' choice to use such highly passaged articular chondrocytes. It is well known that articular chondrocytes are fully differentiated, but it would be a more direct comparison with BM-MSCs if lower passages (P0-1) were used, which retains more biologically relevant profiles. If the relevant assays repeated with low passage articular chondrocytes resulted in the same trends as the authors show, this would strengthen their hypothesis. Although conceded on page 12, line 9, more in-depth justification in this section or the discussion would help defend the authors' methodology.

Author response: We used expanded articular chondrocytes (ACh) for comparison in our study, as this is the best control we could obtain. Importantly, these cells retained articular chondrocyte-like features in that they express the relevant cartilage-associated markers and did not express hypertrophic genes or mineralize (unlike BMSC), making them a suitable cell type for comparison.

We purchased our ACh from a vendor because we had limited access to ACh from young healthy donors. At first thaw, the commercially-obtained ACh were already at passage 3 and required expansion to obtain sufficient numbers for experimentation. It is possible that lower passage ACh may be more responsive to TGF- β . At this point we do not have access to ACh from young, healthy individuals (as we do for the BMSC).

We have added the text below to the discussion (page 17, line 397):

It is possible that lower passage ACh may have a more profound response to short-term TGF- β 1 exposure, compared with high passage ACh, used in our study. However, the number of ACh that can be obtained from healthy individuals is a general limitation, and one of the reasons why BMSCs are being actively investigated as an alternative starting source for generating cartilage tissue.

Referee comment: Inclusion of 1 day of TGF β exposure in Figure 5 would help support the overall hypothesis that it is sufficient to drive differentiation, since the figure begins at 3 days of TGF β exposure.

Author response: We have added to Figure 5, Day 1 TGF- β 1 exposure data for BMSC, including images which show toluidine blue staining and mineralization following *in vivo* incubation. We have modified the ACh panel, showing that, in addition to short (3-day) TGF- β 1 exposure, prolonged (21 day) exposure to TGF- β 1 does not yield hypertrophic/mineralized tissues.

Referee comment: IHC/IFC characterization of tissue formed (i.e., perilipin, alkaline phosphatase, osterix, etc.) in Figure 5 would also be beneficial to definitively observe mineralization detected in the Micro-CT scan, thus supporting their hypothesis that minimal exposure of TGF β is sufficient to drive hypertrophic tissue formation.

Author response: We have added data to Figure 5, showing that 1 day of TGF- β 1 exposure is sufficient to cause mineralization of BMSC *micro*-pellets when incubated *in vivo*. We believe that evidence of hypertrophy and mineralization is clear from our data. Firstly, *in vitro* analysis of gene expression (qPCR and RNA-seq) point to a hypertrophic gene expression signature, then following *in vivo* implantation, the tissues mineralize. It is clear from our data that the *in vivo* implanted tissues are undergoing remodelling into histologically

identifiable bone (as indicated by establishment of bone marrow) and most importantly that mineralization has occurred (evidenced by microCT). We have not performed IHC staining for the markers mentioned by the reviewer. It would be very hard to argue, based on what we have shown, that these tissues are not hypertrophic.

Referee comment: In regards to Figure 6, it would be useful if, instead of describing all theoretically possible factors for hypertrophy regulation, a list of key possible factors that drive hypertrophic mechanisms would be advantageous. With this depicted summary, relevant blockers (WNT and/or BMP blockers) used in vitro could validate the authors' claims of value for identifying these key factors, to ensure the quality of cartilage that one would need for clinical significance.

Author response: We detail what we believe are likely useful targets for mitigating hypertrophy in the discussion. We have already simplified a very long list of potential targets (from our RNA-seq comparisons; BMSC vs ACh) into heatmaps based on genes that have been previously annotated as related to chondrogenesis or osteogenesis. We discuss the targets that have already been trialled with limited success (BMP, WNT signalling), and we do not think that more of the same would be informative. Based on our analysis, and we include this in our discussion, SP7 (Osterix), appears to be a very early transcription factor that could be a good target for mitigating hypertrophy in BMSC. As there are no small molecule inhibitors of SP7, we cannot suggest any logical inhibitors to use at this time. This target will have to be assessed further in future studies, perhaps, initially using gene engineering strategies.

Referee comment: As far as technicalities, Page 7 line 27-28 specifies the use of red arrows in Figure 3a, however there are no red arrows included in the image.

Author response: This has been corrected, thank you.

Referee comment: Figure 4 seems to be a duplicate panel.

Author response: This has been corrected, thank you.

Referee comment: On page 18, line 5, it is assumed that the authors meant "Briefly," rather than "Brief,".

Author response: No, we mean "Brief". As in, the exposure was brief.

Referee comment: On page 21, line 13, it is also assumed that a Greek letter B will be used in lieu of "TGF-B1".

Author response: This has been corrected, thank you.

Referee comment: In general, the authors may want to consider shortening/condensing the Discussion section; it may be more impactful if the overall messages were discussed and presented in a more concise manner.

Author response: We agree that there are a few key points that need to be communicated, but we do want to take advantage of this opportunity to communicate a number of important insights. We considered shortening the text, and trying to communicate insights using an additional schematic/figure, but came to the conclusion that this restricted our capacity to briefly reflect on the literature, and then communicate the new insight(s) embedded in the extensive data set. We hope readers benefit from this detailed discussion.

References

1. Somoza, R.A., et al., *Chondrogenic differentiation of mesenchymal stem cells: challenges and unfulfilled expectations*. Tissue Eng Part B Rev, 2014. **20**(6): p. 596-608.
2. Allen, J.L., M.E. Cooke, and T. Alliston, *ECM stiffness primes the TGFbeta pathway to promote chondrocyte differentiation*. Mol Biol Cell, 2012. **23**(18): p. 3731-42.
3. McMurtrey, R.J., *Analytic Models of Oxygen and Nutrient Diffusion, Metabolism Dynamics, and Architecture Optimization in Three-Dimensional Tissue Constructs with Applications and Insights in Cerebral Organoids*. Tissue Eng Part C Methods, 2016. **22**(3): p. 221-49.
4. Akkerman, N. and L.H. Defize, *Dawn of the organoid era: 3D tissue and organ cultures revolutionize the study of development, disease, and regeneration*. Bioessays, 2017. **39**(4).
5. Futrega, K., et al., *The microwell-mesh: A novel device and protocol for the high throughput manufacturing of cartilage microtissues*. Biomaterials, 2015. **62**: p. 1-12.
6. Markway, B.D., et al., *Enhanced chondrogenic differentiation of human bone marrow-derived mesenchymal stem cells in low oxygen environment micropellet cultures*. Cell Transplant, 2010. **19**(1): p. 29-42.
7. Wang, W., D. Rigueur, and K.M. Lyons, *TGFbeta signaling in cartilage development and maintenance*. Birth Defects Res C Embryo Today, 2014. **102**(1): p. 37-51.
8. Schmitt, B., et al., *BMP2 initiates chondrogenic lineage development of adult human mesenchymal stem cells in high-density culture*. Differentiation, 2003. **71**(9-10): p. 567-77.
9. Occhetta, P., et al., *Developmentally inspired programming of adult human mesenchymal stromal cells toward stable chondrogenesis*. Proc Natl Acad Sci U S A, 2018. **115**(18): p. 4625-4630.

Reviewers' comments:

Reviewer #1 (Remarks to the Author):

Thank you for addressing my concerns and clarifying your methods. You have a very nice contribution to the field.

Reviewer #2 (Remarks to the Author):

The authors did not provide convinced high quality data to answer the following comment "The in vivo part of this study should be supported by more immune-staining for both bone and cartilage specific matrix proteins"

Reviewer #3 (Remarks to the Author):

In its current form, the manuscript does not provide enough supporting data to address my concerns without further experimentation and characterization. While I believe that this manuscript merits publication, the language currently used can be considered as an over-interpretation and should be softened, based on the data presented. Here are some examples on verbiage that needs to be addressed:

- The novelty of a micro-pellet demonstrating more chondrogenic capacity has been described before [1, 2], thus the conceptual novelty this manuscript has to offer is the determination of exposure times with TGFb. The title should also reflect that hypertrophic cartilage is what is intentionally formed, not classical articular cartilage that the authors make direct comparisons with.
- The authors provided no additional data to address the characterization of the in vivo tissue described in Figure 5. Because they have chosen not to provide sufficient evidence to strengthen their claim that the tissue is "bone-marrow like" from the qualitative images, this is an over-interpretation. I suggest they revise their claims to say "bone", which would be supported by the current data.

The authors did not perform nor provide any additional data to address the direct relationship between pellet size and TGFb availability, as the authors allude to many times in their manuscript. Without providing this data, this would be a fundamental flaw in their current claims made. It would be beneficial if the link between micro-pellets and access to TGFb were made more explicitly via experimentation aforementioned, however, changing the claims made would suffice. For instance:

- The authors state that their pilot data with 2D BMSC culture exposed to TGFb for 1 day "failed to generate cartilage-like tissue". This would imply that pelleting the MSCs before TGFb exposure is key for chondrogenesis, or at that hypertrophic characteristics are achieved as early as one day, in conjunction with pelleting. This should be explicitly described early within the manuscript.
- The authors' rebuttal states that they do not claim that "TGFb availability is the major limitation in macro-pellets", but their manuscript heavily opposes this statement (Page 5, lines 17 -18; Page 16, lines 20-22; etc.). Without any further data to support the concept of a smaller pellet having more access to a differentiating factor (TGFb) shown by varying size, or the underlying mechanism of why a micro-pellet produces better cartilage, the arguments made by the authors are insufficient based on the data provided. If the authors explicitly support a relationship between pellet size & TGFb diffusion availability to influence hypertrophic capacity, then their claim that 1 day of TGFb exposure in conjunction with a smaller pellet is sufficient for hypertrophic induction can be supported.

[1] <https://pubmed.ncbi.nlm.nih.gov/19878627/>

[2] <https://pubmed.ncbi.nlm.nih.gov/27699833/>

General: We thank the Reviewers for their comments and suggestions. We have modified the manuscript to include the recommended caveats or to provide clarity. We address each Reviewer comment/question below. Reviewer comments/questions are in black text, and Author response in blue text. New text that has been added to the manuscript is highlighted in yellow, with the page and line number indicating where the additional text has been inserted into the manuscript. This text is also highlighted in yellow in the revised manuscript.

Reviewer 1: Thank you for addressing my concerns and clarifying your methods. You have a very nice contribution to the field.

Authors: Thank you for your contribution to review and improvement of our manuscript.

Reviewer 2: The authors did not provide convinced high quality data to answer the following comment "The *in vivo* part of this study should be supported by more immune-staining for both bone and cartilage specific matrix proteins"

Author Response: In Figure 5, and in the associated text, we specifically focused on the critical insight that all of the BMSC-derived *micro*-pellets mineralized, while none of the ACh *micro*-pellets mineralized. Mineralization of the *micro*-pellets was consistent with the gene expression and RNA-Seq data which demonstrated that all BMSC-derived *micro*-pellets express a hypertrophic gene signature, while none of the ACh *micro*-pellets expressed a hypertrophic gene signature. The process of BMSC hypertrophy, mineralization, and replacement of cartilage-like tissue with marrow is described in previous publications, as cited. This is an extensive remodelling process *in vivo*, in which mineralization occurs early. The microCT scans of the tissues provide a definitive readout for mineralization. We reason that readers will find our manuscript useful specifically because we show that BMSC-derived *micro*-pellets exposed to even a single day of TGF- β 1 *in vitro* formed mineralized tissue *in vivo*.

If the *in vivo* assay were extended, the cartilage-like tissue would likely be progressively replaced by bone and bone marrow. The goal of the assay was to detect mineralization, not to determine the transient quality of the bone-like and bone marrow-like tissue. The quality and quantity of bone-like tissue changes with time, and we would require many time points to characterise this dynamic process. We do not feel that further analysis of the bone quality will add significant value to our manuscript, nor alter the conclusion of our study. Nevertheless, the Reviewer's question identifies an opportunity to improve our discussion through the review of relevant concepts, and historical literature on this topic. Below we provide some context for the new text incorporated into the manuscript.

Background on BMSC hypertrophy: The propensity of chondrogenically induced BMSC to form mineralized tissue *in vivo* has been studied and characterized previously [1-3]. These studies characterized the progressive replacement of the cartilage-like matrix, with mineralized matrix, and vascularized tissue containing bone marrow structures. For example, one study demonstrated that when BMSC tissues were cultured *in vitro* in chondrogenic induction medium, they showed no signs of mineralization [1]. However, approximately three weeks after these chondrogenically induced tissues were implanted *in vivo*, partial mineralization was observed. In non-mineralized areas, the cartilage-like tissue was progressively replaced by vascular stromal tissue [1]. The stromal tissue was progressively populated by mouse hematopoietic cells, and by week 8, marrow structures were obvious within the cores of the remodelled BMSC pellets. In a second study [2], chondrogenically induced BMSC pellets were implanted *in vivo*, and by 4-5 weeks there was extensive calcification of the matrix and vascular invasion that yielded small marrow cavities. A third study implanted chondrogenically induced BMSC pellets in mice for 14 weeks, at which point all cartilage-like tissue had been remodelled and replaced by mineralised bone tissue and marrow stroma [3]

To address the Reviewer's concerns, we have modified a paragraph in the Results section. This paragraph communicates key references and known aspects of BMSC hypertrophic processes, including tissue mineralization, and the progressive replacement of cartilage-like tissue with vascularized tissue and bone marrow.

Page 11, Line 15.

Previous studies show that when BMSC pellets, that have been exposed to TGF- β for 3-7 weeks *in vitro*, are implanted ectopically in mice, that they are iteratively remodelled [1-3]. This remodelling results in mineralization and progressive replacement of cartilage-like tissue with bone or bone marrow-like tissue. Our data demonstrate that BMSC *micro*-pellets exposed to as little as a single day of TGF- β 1 mineralized *in vivo*. Remodelling of *micro*-pellets at the periphery was obvious, and in some cases, small pockets of marrow could be observed. In contrast to BMSC *micro*-pellets, ACh *micro*-pellets did not mineralize *in vivo* (Fig. 5b).

Reviewer 3: The novelty of a micro-pellet demonstrating more chondrogenic capacity has been described before [4, 5], thus the conceptual novelty this manuscript has to offer is the determination of exposure times with TGF β . The title should also reflect that hypertrophic cartilage is what is intentionally formed, not classical articular cartilage that the authors make direct comparisons with.

Author Response: Thank you for this suggestion. The Editors at Communications Biology have generously removed the character limit for the manuscript title. We characterized the behaviour of BMSC in the chondrogenic induction medium formulation developed by Johnstone *et al.* (*In vitro* chondrogenesis of bone marrow-derived mesenchymal progenitor cells [6]), which remains the most commonly used chondrogenic induction medium today. We agree with the Reviewer that the BMSC-derived tissues evolve to be hypertrophic. When BMSC are exposed to a single day of TGF- β 1 this upregulates both chondrogenic and hypertrophic machinery. It is possible that a single day of TGF- β 1 could be used to generate stable cartilage-like tissue, but this would require either chemical or gene modification to dampen parallel hypertrophic gene cascades. We discuss this possibility in the manuscript (Page 18, line 1-24). Our goal with the manuscript title is to provide as much insight as possible, and to attract readers who use TGF- β induction protocols in their efforts to generate cartilage-like tissue from BMSC. With this reasoning in mind, we propose the following manuscript title modification:

The current title of the manuscript is:

Bone marrow stromal cell (BMSC) chondrogenic fate is determined following a single day of TGF- β 1 exposure

The proposed modified manuscript title is:

A single day of TGF- β 1 exposure activates chondrogenic and hypertrophic differentiation pathways in bone marrow-derived stromal cells (BMSC)

Reviewer 3: The authors provided no additional data to address the characterization of the *in vivo* tissue described in Figure 5. Because they have chosen not to provide sufficient evidence to strengthen their claim that the tissue is “bone-marrow like” from the qualitative images, this is an over-interpretation. I suggest they revise their claims to say “bone”, which would be supported by the current data.

Author Response: We have modified the wording in the Results section (Page 11, Line 13, 15-21). As advised, we changed “bone marrow-like” to “bone-like”, and then briefly explained our results in the context of previous literature.

Reviewer 3: The authors did not perform nor provide any additional data to address the direct relationship between pellet size and TGF β availability, as the authors allude to many times in their manuscript. Without providing this data, this would be a fundamental flaw in their current claims made. It would be beneficial if the link between micro-pellets and access to TGF β were made more explicitly via experimentation aforementioned, however, changing the claims made would suffice. For instance:

- The authors state that their pilot data with 2D BMSC culture exposed to TGF β for 1 day “failed to generate cartilage-like tissue”. This would imply that pelleting the MSCs before TGF β exposure is key for chondrogenesis, or at that hypertrophic characteristics are achieved as early as one day, in conjunction with pelleting. This should be explicitly described early within the manuscript.

Author Response: Pilot data provided during the first peer review show that the addition of TGF- β 1 to BMSC expansion medium did not improve outcomes, nor replicate the effects of brief TGF- β 1 exposure on BMSC that had been aggregated into *micro*-pellets. To put these studies into the main paper would require that they be replicated across a number of cell populations, cell densities, medium formulations and for different durations of TGF- β 1 exposure. As this approach does not appear promising, and because it is not the focus of this manuscript, we do not propose to pursue this method further. The peer review discussion will be published by Communications Biology, hopefully providing interested parties with our detailed dialogue. Nevertheless, Reviewer 3 is correct to highlight the importance of aggregation in this process. *In vivo* mesenchymal stem cell aggregation (condensation) is requisite for cartilage formation [7]. *In vitro*, TGF- β 1 coordinates chondrogenic differentiation in part through the indirect down regulation of N-cadherin in BMSC pellets over the first day of culture [8]. This down regulation of N-cadherin contributes to a relaxation

of cytoskeletal tension, and RhoA/ROCK signaling. Pharmacological inhibition of ROCK similarly contributes to reduced cytoskeletal tension, yielding improved chondrogenic differentiation [9, 10], and upregulation of Sox9. By contrast, over expression of RhoA downregulates these chondrogenic processes. In addition to the physical cues provided by cell aggregation, the geometry of cell aggregates is superior to a monolayer in that it readily accommodates matrix accumulation. Cells in aggregates can deposit matrix on all sides, while deposition of matrix on all sides of a monolayer culture can confound results by causing delamination of the monolayer. The above benefits are some of the reasons that pellet cultures are routinely used to study BMSC chondrogenesis. To improve the clarity of the manuscript, and to address Reviewer 3's comments we have added the following text:

Page 16, Line 1-5

Pelleted BMSC are a logical input into the chondrogenic differentiation process. *In vivo* mesenchymal stem cell aggregation (condensation) is requisite for cartilage formation [7], while *in vitro* TGF- β 1 facilitates BMSC chondrogenic differentiation in part through the indirect down regulation of N-cadherin, causing relaxation of cytoskeletal tension leading to reduced RhoA/ROCK signaling and upregulation of chondrogenic signalling [8].

Reviewer 3: The authors' rebuttal states that they do not claim that "TGF β availability is the major limitation in macro-pellets", but their manuscript heavily opposes this statement (Page 5, lines 17 -18; Page 16, lines 20-22; etc.). Without any further data to support the concept of a smaller pellet having more access to a differentiating factor (TGF β) shown by varying size, or the underlying mechanism of why a micro-pellet produces better cartilage, the arguments made by the authors are insufficient based on the data provided. If the authors explicitly support a relationship between pellet size & TGF β diffusion availability to influence hypertrophic capacity, then their claim that 1 day of TGF β exposure in conjunction with a smaller pellet is sufficient for hypertrophic induction can be supported.

Author Response: Many apologies for any confusion. In the referenced sentences (Page 5, lines 17 -18; Page 16, lines 20-22; etc.) we did not explicitly state that the relevant gradients were of TGF- β 1. Rather, many metabolite and signal gradients form through the diameter of a pellet, and these gradients will be more profound in larger pellets. It is true that in our system a TGF- β 1 gradient is not decoupled from other metabolite or signal gradients. It is likely that the reduction in multiple diffusion gradients in micro-pellet contribute to the observed outcomes. Thus, we prefer the explanation that when diffusion gradients (all gradients) are minimized, that it becomes apparent that a single day of TGF- β 1 exposure is sufficient to trigger BMSC differentiation cascades. Despite using classical chondrogenic induction medium [6], BMSC in both *macro*- or *micro*-pellets engage both chondrogenic and hypertrophic differentiation processes. **The critical observation described herein is that if signals/metabolites are not limited by diffusion, a single day of TGF- β 1 exposure is sufficient to activate both chondrogenic and hypertrophic machinery.** To address Reviewer 3's question "smaller pellet is sufficient for hypertrophic induction": The smaller *micro*-pellet and brief TGF- β 1 exposure is sufficient for both chondrogenic and hypertrophic induction. Reviewer 3 is correct in noting that this sequence of events ultimately yields tissue where hypertrophic features dominate. To make clear that many gradients likely contribute to outcomes, and not specifically a TGF- β 1 gradient, we have modified the following text in the manuscript:

Page 5, line 14-19

Because individual *micro*-pellets (5×10^3 cells each) are formed from fewer cells than *macro*-pellets (2×10^5 cells each), the resulting smaller tissues experience reduced diffusion gradients, generally leading to the more uniform supply of both metabolites and signal molecules. The resulting more homogeneous cartilage-like tissues are better suited for studying BMSC chondrogenic differentiation kinetics.

and

Page 16, line 26-28

A range of metabolite and signal gradients across the diameter of *macro*-pellets likely contributed to this heterogeneity.

References

1. Serafini, M., B. Sacchetti, A. Pievani, D. Redaelli, C. Remoli, A. Biondi, M. Riminucci, and P. Bianco, *Establishment of bone marrow and hematopoietic niches in vivo by reversion of chondrocyte differentiation of human bone marrow stromal cells*. Stem Cell Res, **2014**. 12(3): p. 659-72.
2. Pelttari, K., A. Winter, E. Steck, K. Goetzke, T. Hennig, B.G. Ochs, T. Aigner, and W. Richter, *Premature induction of hypertrophy during in vitro chondrogenesis of human mesenchymal stem cells correlates with calcification and vascular invasion after ectopic transplantation in SCID mice*. Arthritis Rheum, **2006**. 54(10): p. 3254-66.
3. Farrell, E., S.K. Both, K.I. Odorfer, W. Koevoet, N. Kops, F.J. O'Brien, R.J. Baatenburg de Jong, J.A. Verhaar, V. Cuijpers, J. Jansen, R.G. Erben, and G.J. van Osch, *In-vivo generation of bone via endochondral ossification by in-vitro chondrogenic priming of adult human and rat mesenchymal stem cells*. BMC Musculoskelet Disord, **2011**. 12: p. 31.
4. Markway, B.D., G.K. Tan, G. Brooke, J.E. Hudson, J.J. Cooper-White, and M.R. Doran, *Enhanced chondrogenic differentiation of human bone marrow-derived mesenchymal stem cells in low oxygen environment micropellet cultures*. Cell Transplant, **2010**. 19(1): p. 29-42.
5. Ouyang, A., A.E. Cerchiari, X. Tang, E. Liebenberg, T. Alliston, Z.J. Gartner, and J.C. Lotz, *Effects of cell type and configuration on anabolic and catabolic activity in 3D co-culture of mesenchymal stem cells and nucleus pulposus cells*. J Orthop Res, **2017**. 35(1): p. 61-73.
6. Johnstone, B., T.M. Hering, A.I. Caplan, V.M. Goldberg, and J.U. Yoo, *In vitro chondrogenesis of bone marrow-derived mesenchymal progenitor cells*. Exp Cell Res, **1998**. 238(1): p. 265-72.
7. Hall, B.K. and T. Miyake, *Divide, accumulate, differentiate: cell condensation in skeletal development revisited*. Int J Dev Biol, **1995**. 39(6): p. 881-93.
8. Tuli, R., S. Tuli, S. Nandi, X. Huang, P.A. Manner, W.J. Hozack, K.G. Danielson, D.J. Hall, and R.S. Tuan, *Transforming Growth Factor- β -mediated Chondrogenesis of Human Mesenchymal Progenitor Cells Involves N-cadherin and Mitogen-activated Protein Kinase and Wnt Signaling Cross-talk*. Journal of Biological Chemistry, **2003**. 278(42): p. 41227-41236.
9. Wang, K.-C., T.T. Egelhoff, A.I. Caplan, J.F. Welter, and H. Baskaran, *ROCK Inhibition Promotes the Development of Chondrogenic Tissue by Improved Mass Transport*. Tissue Engineering Part A, **2018**. 24(15-16): p. 1218-1227.
10. Woods, A., G. Wang, and F. Beier, *RhoA/ROCK Signaling Regulates Sox9 Expression and Actin Organization during Chondrogenesis*. Journal of Biological Chemistry, **2005**. 280(12): p. 11626-11634.

REVIEWERS' COMMENTS:

Reviewer #2 (Remarks to the Author):

The authors modified the results as requested

Reviewer #3 (Remarks to the Author):

The authors have changed the title to better reflect the outcomes of their study, which help mitigate some potential confusion about the results. However, they did not sufficiently address the concerns both Reviewer 2 and I have about characterization of the in vivo tissue generated. Their revised text on Page 11, line 20 still overstates the observation of "small pockets of [bone] marrow", without providing the necessary evidence to support these claims; only citing references.

Furthermore, the authors failed to address the main concern regarding TGFb1 availability and pellet size. The text added on Page 16, line 1-5, is insufficient to address their pilot data which suggests that it is not just TGFb exposure that drives hypertrophic chondrocyte gene expression, but also with conjunction of pelleting. The text added on Page 16 does not address that their pilot data showed TGFb1 exposure in 2D did not improve outcomes, and that it is concurrent TGFb exposure AND pelleting that drive chondrogenic gene expression -- not exclusively TGFb availability. With no generation of new data that add support to these claims about smaller pellet size and TGFb or "signal molecule" availability, as outlined in the first revision dialogue, this manuscript is not suitable for publication at this time.

Text revisions are not sufficient enough to address my concerns; however the reviewer understands that some limitations and lack of new data may be related to the current situation with COVID-19 pandemic

We are grateful for the Reviewers' contribution to improving our manuscript. Reviewer comments are in black text below. Our response is in blue text. Changes in the manuscript are highlighted.

REVIEWERS' COMMENTS:

Reviewer 2 comment: The authors modified the results as requested

Author response: Thank you for your contributions to this paper.

Reviewer 3 comment: The authors have changed the title to better reflect the outcomes of their study, which help mitigate some potential confusion about the results.

Author response: Thank you.

Reviewer 3 comment: However, they did not sufficiently address the concerns both Reviewer 2 and I have about characterization of the in vivo tissue generated. Their revised text on Page 11, line 20 still overstates the observation of "small pockets of [bone] marrow", without providing the necessary evidence to support these claims; only citing references.

Author response: We are sorry that this was not clear and have modified the text, specifically pointing to regions between BMSC *micro*-pellets in Figure 5a (page 11, line 21); these representative images have a 'bone marrow-like' histological appearance based on the sinusoid-like structures surrounded by hematopoietic cells.

"Remodelling of *micro*-pellets at the periphery was obvious, and in some cases, small pockets of marrow could be observed (evident between BMSC *micro*-pellets exposed to TGF- β 1 for 21 days in Fig. 5a)."

Strikingly, this development of bone marrow-like tissue is not present in any histological sections of articular chondrocyte-derived tissues we analysed because they do not recruit the necessary cells to form these features.

Reviewer 3 comment: Furthermore, the authors failed to address the main concern regarding TGF β 1 availability and pellet size. The text added on Page 16, line 1-5, is insufficient to address their pilot data which suggests that it is not just TGF β exposure that drives hypertrophic chondrocyte gene expression, but also with conjunction of pelleting. The text added on Page 16 does not address that their pilot data showed TGF β 1 exposure in 2D did not improve outcomes, and that it is concurrent TGF β exposure AND pelleting that drive chondrogenic gene expression -- not exclusively TGF β availability. With no generation of new data that add support to these claims about smaller pellet size and TGF β or "signal molecule" availability, as outlined in the first revision dialogue, this manuscript is not suitable for publication at this time. Text revisions are not sufficient enough to address my concerns; however the reviewer understands that some limitations and lack of new data may be related to the current situation with COVID-19 pandemic

Author response: We have modified the text to describe the logic of pelleting, and the limitations (page 16). We are cautious not to overstate the necessity of 3D culture (pelleting), as there are many permutations that should be trialled before excluding the possibility that BMSC monolayer exposure to TGF- β 1 is not effective. For example, other

medium components or cell density are likely to influence BMSC response to TGF- β 1 in monolayer culture. A higher cell density or medium supplementation with a Rho kinase (ROCK) inhibitor could mimic the benefits of cell pelleting.